**EMBO** *reports*

# Ageing-associated long non-coding RNA extends lifespan and reduces translation in non-dividing cells

Shajahan Anver [1], Ahmed Faisal Sumit [1], Xi-Ming Sun[2,3], Abubakar Hatimy[4],
Konstantinos Thalassinos [4,5], Samuel Marguerat [2,3,6], Nazif Alic[1] & Jürg Bähler [1✉]

## Abstract

**Genomes produce widespread long non-coding RNAs (lncRNAs) of largely unknown functions. We characterize *aal1* (ageing-associated lncRNA), which is induced in quiescent fission yeast cells. Deletion of *aal1* shortens the chronological lifespan of non-dividing cells, while ectopic overexpression prolongs their lifespan, indicating that *aal1* acts in trans. Overexpression of *aal1* represses ribosomal-protein gene expression and inhibits cell growth, and *aal1* genetically interacts with coding genes functioning in protein translation. The *aal1* lncRNA localizes to the cytoplasm and associates with ribosomes. Notably, *aal1* overexpression decreases the cellular ribosome content and inhibits protein translation. The *aal1* lncRNA binds to the *rpl1901* mRNA, encoding a ribosomal protein. The rpl1901 levels are reduced ~2-fold by aal1, which is sufficient to extend lifespan. Remarkably, the expression of the *aal1* lncRNA in *Drosophila* boosts fly lifespan. We propose that *aal1* reduces the ribosome content by decreasing Rpl1901 levels, thus attenuating the translational capacity and promoting longevity. Although *aal1* is not conserved, its effect in flies suggests that animals feature related mechanisms that modulate ageing, based on the conserved translational machinery.**

**Keywords** Protein Translation; Chronological Lifespan;
*Schizosaccharomyces pombe*; Ribosomal Protein; RNA Regulation
**Subject Categories** RNA Biology; Translation & Protein Quality

## Introduction

Ageing is a complex, multifactorial process leading to a gradual decline in biological function over time (Lopez-Otin et al, 2023). Old age is a shared root cause for complex diseases such as cancer, neurodegeneration, and cardiovascular or metabolic disorders (Guo et al, 2022). Ageing is highly plastic: simple genetic, nutritional, or pharmacological interventions in model organisms can extend lifespan (Niccoli and Partridge, 2012). But a major challenge for ageing research is understanding all factors affecting lifespan. Ageing-related processes are remarkably conserved from yeast to humans, with simple, short-lived model organisms remaining the main platform to discover and dissect factors that modulate ageing processes (Fontana et al, 2010).

Using fission yeast (*Schizosaccharomyces pombe*) as a simple model for cellular ageing, we and others have analysed genetic and environmental factors determining chronological lifespan (CLS) (Rallis et al, 2014; Romila et al, 2021; Roux et al, 2009; Sideri et al, 2014). CLS is a complex trait defined as the time cells survive under limiting nutrients in a non-dividing state, which enlightens the ageing of post-mitotic and quiescent mammalian cells (Fontana et al, 2010). Quiescence is characterized by a reversible arrest of cell proliferation, increased stress resistance, and reprogramming of gene expression and metabolism from a growth mode to a maintenance mode (De Virgilio, 2012; Marguerat et al, 2012; Roche et al, 2017; Valcourt et al, 2012). Quiescence is a highly prevalent yet under-studied state relevant to cellular and organismal ageing. Quiescent states like dauer worms and non-dividing yeast cells have revealed universal ageing-related processes, e.g. the conserved TORC1 nutrient-signalling network that controls quiescence entry and longevity from yeast to mammals by regulating growth, metabolism, and protein translation (De Virgilio, 2012; Fontana et al, 2010). Human cells alternating between cellular quiescence and proliferation are critical for ageing- and disease-associated processes, including stem-cell function, tissue homeostasis/renewal, immune responses, and drug resistance of tumours (Li and Clevers, 2010; Oh et al, 2014; Valcourt et al, 2012). The ageing and depletion of quiescent stem cells may be an important driver of organismal ageing (Oh et al, 2014; Ren et al, 2017).

Genomes are pervasively expressed, e.g. about 75% of the human genome is transcribed yet less than 2% codes for proteins. A substantial portion of transcriptomes consists of long non-coding RNAs (lncRNAs), which are longer than 200 nucleotides, do not overlap any coding RNAs, and can play varied roles in gene regulation at multiple levels (Herman et al, 2022; Hon et al, 2017; Tsagakis et al, 2020; Ulitsky and Bartel, 2013). Functional analyses of lncRNAs are challenging owing to poor annotation, low

[1]Institute of Healthy Ageing, Research Department of Genetics, Evolution and Environment, University College London, London WC1E 6BT, UK. [2]Institute of Clinical Sciences, Imperial College London, London W12 0NN, UK. [3]MRC London Institute of Medical Sciences (LMS), London W12 0NN, UK. [4]Institute of Structural and Molecular Biology, Division of Biosciences, University College London, London WC1E 6BT, UK. [5]Institute of Structural and Molecular Biology, Birkbeck College, University of London, London WC1E 7HX, UK. [6]Present address: UCL Cancer Institute, University College London, London WC1E 6BT, UK. ✉E-mail: j.bahler@ucl.ac.uk

expression, and limited methodology (Gao et al, 2020; Kopp and Mendell, 2018; Mattick et al, 2023; Ponting and Haerty, 2022). Knowledge of lncRNAs is therefore far from complete even in well-studied organisms. Although lncRNAs show little sequence conservation, functional mechanisms and interacting proteins may be conserved (Herman et al, 2022; Ulitsky and Bartel, 2013).

Mounting evidence implicates certain lncRNAs functioning in ageing and associated diseases (Abraham et al, 2017; Ghafouri-Fard et al, 2022; Grammatikakis et al, 2014; Jiang et al, 2021; Kour and Rath, 2016; Ni et al, 2022; Sherazi et al, 2023; So et al, 2022). For example, lncRNAs play vital roles in ageing-associated NFκB signalling (Cai and Han, 2021) and many lncRNAs are differentially expressed in ageing human fibroblasts, e.g. lncRNA1 which delays senescence (Abdelmohsen et al, 2013). Specific lncRNAs are biomarkers for age-associated diseases and could provide more readily accessible drug targets than proteins (Bravo-Vazquez et al, 2023; Tsagakis et al, 2020; Zhou et al, 2019). Hence, lncRNAs are emerging as important yet poorly characterized ageing factors, presenting a promising research frontier. Fission yeast, featuring an RNA metabolism similar to metazoa, provides a powerful model system and living test tube to study lncRNA function (Atkinson et al, 2018; Fauquenoy et al, 2018; Ono et al, 2022; Rodriguez-Lopez et al, 2022).

Recently, we have reported cellular phenotypes for 150 *S. pombe* lncRNA mutants in over 150 different nutrient, drug, and stress conditions (Rodriguez-Lopez et al, 2022). Phenotype correlations revealed a cluster of four lncRNAs with roles in meiotic differentiation and the survival of quiescent spores. Here, we characterize one of these lncRNAs, *SPNCRNA.1530*, and show that it extends the CLS, interacts with ribosomal proteins, and reduces the ribosome content and cell growth. We name this lncRNA *aal1* for *ageing-associated lncRNA1*. Remarkably, *aal1* expression in *Drosophila* is sufficient to extend the lifespan of flies, suggesting that the functional principle through which *aal1* acts is conserved in animals.

# Results

## *aal1* prolongs the chronological lifespan of non-dividing cells and reduces the growth of proliferating cells

The *aal1* gene (*SPNCRNA.1530*) encodes a 783 nucleotide lncRNA on Chromosome II. The *aal1* transcript levels are induced ~6–24-fold in RNAi, nuclear-exosome, and other RNA-processing mutants (Fig. 1A) (Atkinson et al, 2018), suggesting that these RNA-processing pathways actively degrade *aal1* in proliferating cells. As is typical for lncRNAs, *aal1* is expressed below 1 RNA copy/cell on average in proliferating cells (Marguerat et al, 2012), but its transcript levels are induced ~6–30-fold in non-dividing cells, including stationary-phase cells, which are limited for glucose, as well as quiescent and meiotically differentiating cells, which are limited for nitrogen (Fig. 1A). To validate these findings, we applied strand-specific RT-qPCR which showed that the *aal1* transcript levels are induced >15-fold in stationary-phase cells relative to proliferating cells (Fig. 1B). After induction, the cycle threshold (Ct) was 26.9 ± 0.8, reflecting substantial *aal1* transcript levels in non-dividing cells.

We tested the effects of *aal1* on the CLS of non-dividing cells. To this end, we constructed two strains. The *aal1-pOE* strain allows for strong ectopic overexpression of *aal1* from a plasmid under the thiamine-repressible *P41nmt1* promoter (Fig. EV1A), while the *aal1Δ* strain features a complete deletion of the *aal1* gene. Notably, *aal1-pOE* cells showed a prolonged lifespan during stationary phase, whereas *aal1Δ* cells showed a shortened lifespan (Fig. 1C). Thus, *aal1* exerts an anti-ageing, longevity effect in stationary phase cells.

The CLS is often inversely correlated with cell growth (Rallis et al, 2014). Therefore, we tested whether *aal1* affects the growth of proliferating cells. Indeed, *aal1-pOE* cells featured longer lag periods and slower growth rates than control cells (Fig. 1D). Conversely, *aal1Δ* cells showed only a subtly reduced lag period (Fig. 1E). This weak positive growth effect in proliferating *aal1Δ* cells is consistent with *aal1* being hardly expressed in these cells anyway (Marguerat et al, 2012). We conclude that *aal1* has an anti-growth effect in proliferating cells.

While the *aal1* gene locus does not overlap any protein-coding genes, another lncRNA (*SPNCRNA.401*) is located antisense to *aal1* with over 90% sequence overlap (Fig. EV1B). This setup renders it impossible to distinguish between deletions of *aal1* and *SPNCRNA.401*, raising the possibility that the absence of *SPNCRNA.401* could cause phenotypes observed in *aal1Δ* cells. Like *aal1*, *SPNCRNA.401* is induced in non-dividing cells (Atkinson et al, 2018). We, therefore, checked whether overexpression of *SPNCRNA.401* might show effects on lifespan and growth similar to overexpression of *aal1*. As for *aal1*, we constructed a strain for ectopic overexpression (*SPNCRNA.401-pOE*). In contrast to *aal1-pOE* cells, the *SPNCRNA.401-pOE* cells did not significantly affect lifespan or cell growth (Fig. EV1C). If anything, *SPNCRNA.401-pOE* cells showed inverse, but much weaker phenotypes than *aal1-pOE* cells, marginally decreasing lifespan but increasing cell growth. Moreover, ectopic overexpression of *aal1* was sufficient to rescue the short lifespan of *aal1Δ* cells, further supporting that the CLS phenotype reflects the absence of *aal1* (Fig. EV1D). Given that *aal1* is close to the promoters of the flanking coding genes *ckb2* and *ctu1* (Fig. EV1B; encoding a CK2 regulatory subunit and a cytosolic thiouridylase subunit, respectively) and some lncRNAs regulate the expression of neighbouring genes *in cis* (Ard et al, 2014; Hirota et al, 2008; Shah et al, 2014), we also quantified the changes in transcript levels of these neighbouring genes by qRT-PCR. These experiments showed that deletion of *aal1* did, at most, marginally affect the expression of the adjacent genes (Fig. EV1E). Taken together, we conclude that the phenotypes observed in *aal1Δ* and *aal1-pOE* cells reflect the absence and overexpression, respectively, of the *aal1* RNA.

Inhibition of the conserved Target of Rapamycin Complex 1 (TORC1) signalling network leads to decreased cell growth and prolonged lifespan from yeast to mammals (Liu and Sabatini, 2020; Rallis et al, 2013). The *aal1* overexpression phenotypes resemble those of TORC1 inhibition, raising the possibility that *aal1* functions within the TORC1 network. Therefore, we tested whether the *aal1* growth phenotype depends on TORC1 function. TORC1 inhibition by rapamycin and caffeine led to similar relative growth reduction in *aal1-pOE* and *aal1Δ* cells as in the respective controls (Fig. EV2). These results indicate that overexpression of *aal1* and inhibition of TORC1 exert additive effects on cell growth, suggesting that *aal1* acts independently of the TORC1 network.

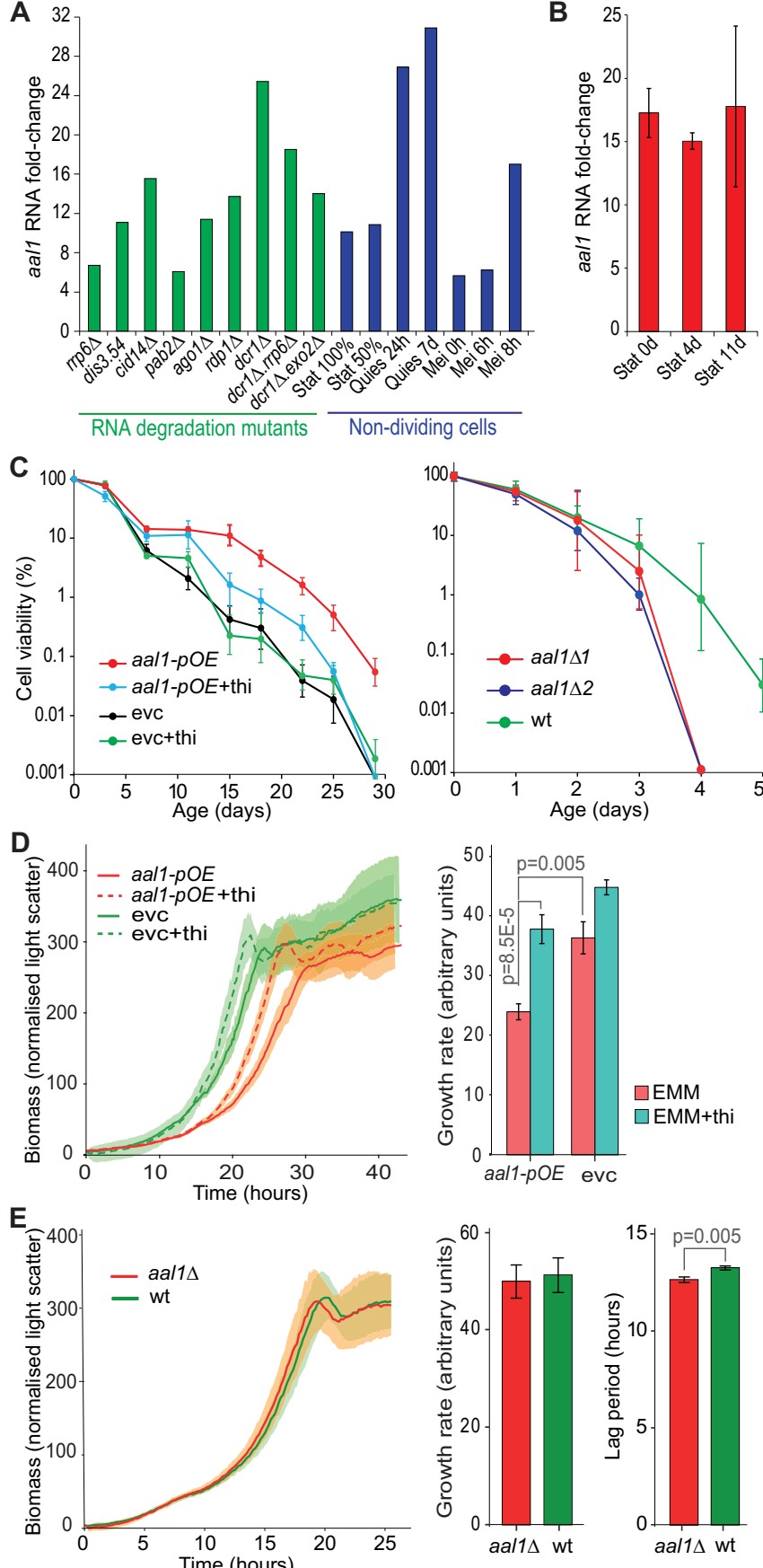

**Figure 1.** *aal1* prolongs the chronological lifespan of non-dividing cells and reduces growth of proliferating cells.

(A) RNA fold-changes for *aal1* in different genetic and physiological conditions relative to wild-type proliferating cells, based on RNA-seq data (Atkinson et al, 2018). Green: various RNA degradation mutants as indicated; blue: various non-dividing cells, including stationary-phase cells (Stat) at 100% and 50% viabilities, quiescent cells (Quies) after 24 h and 7 days, and meiotically differentiating cells (Mei) at 0, 6, and 8 h. (B) RNA fold-changes (log$_2$) for *aal1* at the onset of stationary phase (Stat 0d) and after 4 and 11 days in stationary phase (Stat 4d and 11d) relative to proliferating cells, based on strand-specific RT-qPCR with gene-specific primers (ΔΔCt method). The *aal1* RNA levels are normalized to the lowly expressed coding gene *ppb1*. Bars indicate the means ± standard errors of 3 independent repeats. (C) Left graph: Chronological lifespan assays for cells ectopically overexpressing *aal1* from a plasmid under the *P41nmt1* promoter (*aal1-pOE*) compared to empty-vector control cells (evc). Cells were cultured in the absence or presence of 15 μM thiamine (thi), where the *P41nmt1* promoter is active or repressed, respectively. Right graph: Chronological lifespan assays for *aal1* deletion cells (*aal1Δ*) compared to wild-type cells (wt). Two independent deletion strains (*aal1Δ1* and *aal1Δ2*) were generated by CRISPR-Cas9 using distinct sgRNAs. The percentages of viable cells were measured using a robotics-based colony-forming units (CFUs) assay (Romila et al, 2021). A Poisson distribution-based model was used for maximum likelihood estimates of the number of CFUs and shown in the y-axes as percentages relative to the CFUs at Day 0. Data points reflect the mean ± SE of three biological repeats. Overexpression experiments were performed in minimal medium, while deletion experiments were performed in rich medium. (D) Left graph: Ectopic *aal1* overexpression under the thiamine-repressible *P41nmt1* promoter (*aal1-pOE*) prolongs the lag period and reduces the growth rate compared to empty-vector control (evc) and/or in the absence of 15 μM thiamine (thi). Cells were grown in a microbioreactor, and mean growth curves were fitted with *grofit* (Kahm et al, 2010), with SD from six independent repeats shown as shades. Right graph: Quantitation of growth rate for experiments shown in the left graph. Growth rate calculations were done using *grofit* (Kahm et al, 2010) and statistical significance was determined with a one-way ANOVA followed by Tukey's Honest Significant Difference test for pairwise comparisons in R (R Core Team, 2020). Thiamine generally promotes growth in yeast cells (Sabatinos and Forsburg, 2010), as also observed here. (E) Left graph: Cells deleted for *aal1* (*aal1Δ*) show a slightly decreased lag period but similar growth rate to wild-type cells (wt). Same experimental setup and analysis as in (D). Right graphs: Quantitation of growth rates and lag periods for experiment in the left graph. Statistical significance was determined with one-way ANOVA followed by Dunnett's test (*multcomp*) (Hothorn et al, 2008) to correct for multiple testing of the comparisons of the lag periods and growth rates of *aal1Δ* against wt. Source data are available online for this figure.

We conclude that the expression of the *aal1* lncRNA strongly increases in non-dividing cells, and the absence or excess of the *aal1* RNA is sufficient to shorten or extend the lifespan of these cells, respectively. Moreover, expression of *aal1* in proliferating cells leads to reduced growth. The lifespan and growth phenotypes are mediated by *aal1* independently of the TORC1 network. Notably, *aal1* can promote longevity and repress growth when expressed from a plasmid, indicating that it acts in trans as a lncRNA.

## *aal1* genetically interacts with coding genes functioning in protein translation and localises to the cytoplasm

To obtain clues on the molecular function of *aal1*, we systematically assayed genetic interactions between the non-coding *aal1* gene and protein-coding genes. Combining two mutations in the same cell can cause phenotypes that are more or less severe than expected from the phenotypes of the single mutations, defining negative and positive genetic interactions, respectively. Such genetic (or epistatic) interactions reveal broad relationships between functional modules or biological processes (Bellay et al, 2011). To systematically uncover functional relationships between the *aal1* RNA and proteins, we used an *aal1Δ* strain (*aal1::natMX6*) as the query mutant to screen for interactions with all 3420 non-essential coding-gene deletions with the synthetic genetic array (SGA) method(Baryshnikova et al, 2010; Costanzo et al, 2019). We measured all pairwise genetic interactions using colony size as a proxy for double-mutant fitness relative to *ade6Δ* control double mutants. As for screens with coding-gene query mutants (Rallis et al, 2014; Rallis et al, 2017), we observed moderate reproducibility between the three repeats and more negative than positive interactions, revealing 140 negative and 28 positive genetic interactions for at least two of the three repeats (Dataset EV1). Analysis using AnGeLi (Bitton et al, 2015) showed that these interacting genes were enriched in those showing lifespan and growth phenotypes, as *aal1* itself, including the fission yeast phenotype ontology (FYPO) terms (Harris et al, 2013; Romila et al, 2021; Sideri et al, 2014) 'loss of viability in stationary phase'

($p_{adj}$ = 5.3E−17) and 'abnormal vegetative cell population growth' ($p_{adj}$ = 5.0E−14). Moreover, the interacting genes were strongly enriched for several Gene Ontology (GO) terms (Gene Ontology, 2021) related to ribosome biogenesis/function and cytoplasmic translation (Figs. 2A and EV3). For a complementary assay, we also generated a strain overexpressing *aal1* from the *P41nmt1* promoter at its native genomic locus (*aal1-gOE*), and screened this query mutant for interactions with all non-essential coding-gene deletions. The 297 genes interacting with the *aal1-gOE* mutant were also enriched in several GO terms related to ribosomes and translation, besides those related to metabolism, mitochondrial function and cell regulation (Fig. EV3; Dataset EV1). We conclude that genetic interactions for both *aal1* deletion and overexpression mutants point to functions associated with protein translation.

To further test the possibility that *aal1* functions in translation, we analysed the subcellular localization of the *aal1* RNA using antisense probes for single-molecule RNA fluorescence in situ hybridization (smRNA-FISH) (Sun et al, 2020). Given that the *aal1* RNA expression is very low in proliferating cells (Fig. 1A,B) and smRNA-FISH is challenging in stationary-phase cells (Ellis et al, 2021), we generated a strain expressing *aal1* from the *P41nmt1* promoter at its native genomic locus (*aal1-gOE*). This analysis revealed that *aal1* RNAs predominantly localize in the cytoplasm in multiple foci (Fig. 2B). Similar results were obtained with the *aal1-pOE* strain that ectopically overexpresses *aal1* (Fig. 2B). This finding corroborates that *aal1* functions in trans as a lncRNA and is consistent with a role of *aal1* in cytoplasmic translation.

## *aal1* is associated with ribosomes and reduces the translational capacity

Many lncRNAs function with RNA-binding proteins (RBPs), and identifying such RBPs can shed light on the molecular roles of the target lncRNAs (Faoro and Ataide, 2014). To uncover RBPs that interact with *aal1*, we applied an RNA-centric approach, 'comprehensive identification of RBPs by mass spectrometry' (ChIRP-MS) (Chu et al, 2015). We designed biotinylated antisense oligos along the length of *aal1* to pull down RBPs associated with *aal1*, which

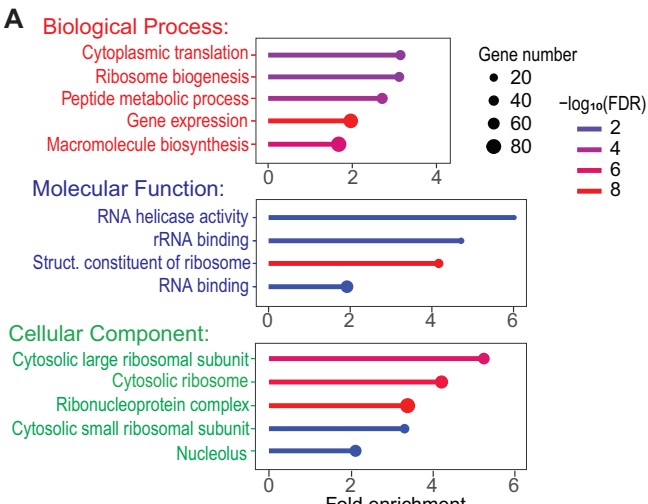

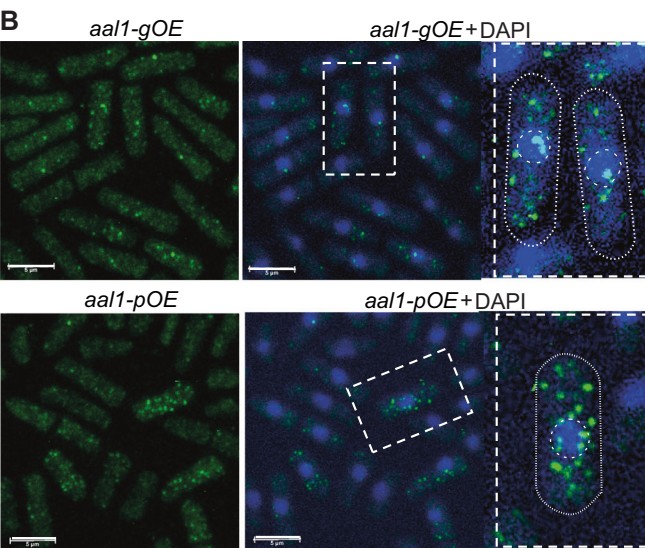

**Figure 2. aal1 shows functional relationships with translation-related processes and localizes to the cytoplasm.**

(A) GO-terms enriched among the genes that showed positive or negative genetic interactions (FDR ≤ 0.05) in at least 2 of the 3 repeats in the SGA screens using aal1Δ as query mutant. Representative GO terms for Biological Process (red), Molecular Function (blue), and Cellular Component (green) are shown, selected for non-redundancy, specificity, and significance. The graphs show the fold enrichments as well as the gene numbers and −log₁₀ of false-discovery rate as indicated in the legends at right. Visualisation with ShinyGO (ver 0.77) (Ge et al, 2020). The genetic-interaction and background gene lists are provided in Dataset EV1. (B) Fluorescence micrographs of single-molecule FISH experiments of aal1-gOE cells (top) and aal1-pOE cells (bottom). The aal1 RNAs are labelled in green and DAPI-stained DNA is shown in blue. Enlarged cells with nuclei are outlined at right. Scale bars: 5 μm. Source data are available online for this figure.

does not require genetic manipulations that could interfere with aal1 function (Methods). We performed ChIRP-MS with wild-type and aal1-pOE cells after 6 days in stationary-phase, when the aal1 transcript levels are induced, with aal1Δ cells serving as control. In total, we identified 218 proteins across all three independent repeats in all strains, 68 of which were significantly more abundant

in the pull-downs from wild-type and/or aal1-pOE cells than from aal1Δ cells (FDR ≤ 0.005 and log₂FC ≥ 6.5; Dataset EV2). These 68 aal1-bound proteins were enriched for biological processes associated with cytoplasmic translation and energy metabolism (Fig. 3A). The 68 proteins were more highly connected with each other than expected by chance for known protein–protein interactions, reflecting an extensive network of ribosomal and ribosome-associated proteins (Fig. 3B). While 27 of the 68 proteins have known functions in translation-related processes, most of the remaining proteins are enzymes functioning in energy metabolism. Recent research indicates that many metabolic enzymes exert moonlighting functions as RBPs that can act as ribosome-associated proteins (Kilchert et al, 2020; Simsek et al, 2017). Notably, 44 of the 68 aal1-bound proteins have been independently shown to be RBPs in S. pombe (Dataset EV2) (Kilchert et al, 2020). Taken together, these results suggest that aal1 directly or indirectly functions with multiple proteins associated with ribosomes and/or ribosomal subunits.

To corroborate and elucidate this ribosomal association, we applied polysome fractionation (Pospisek and Valasek, 2013) followed by RT-qPCR analysis (Bachand et al, 2006). Most aal1 RNA was detected together with free ribosomal subunits (40S/60S) and monosomes (80S) in proliferating aal1-pOE cells (Fig. 3C). As a control mRNA, we used ppb1, the least variable gene under many perturbations, including stationary phase, and is comparatively lowly expressed (Pancaldi et al, 2010). As expected, ppb1 was predominantly found in polysomes (Fig. 3C). Similar aal1 localization patterns were apparent for both wild-type and aal1-pOE cells in young and old stationary-phase cells, although in the older cells, a substantial proportion of aal1 was also detected in polysomes (Fig. EV4A).

We then checked for any effects of aal1 on the polysome profiles. Remarkably, proliferating aal1-pOE cells showed a decreased polysome profile compared to empty-vector control cells (Fig. 4A), while proliferating aal1Δ cells showed an increased profile compared to wild-type control cells (Fig. 4B). These differences were most pronounced for the monosome fractions. Similar patterns were evident in the polysome profiles from aal1-pOE and aal1Δ cells during early and late stationary phase (Fig. EV4B). These results suggest that aal1 has an inhibitory effect on the cellular ribosome content. To establish that the absence of aal1 caused the increased ribosome content in aal1Δ cells, we ectopically expressed aal1 (aal1-pOE) or an empty-vector control in aal1Δ cells. Indeed, ectopic expression of aal1 was sufficient to decrease the ribosome content in aal1Δ cells to levels similar to or below those in wild-type cells (Figs. 4C and EV4C).

To quantify and validate the observed differences in the polysome profiles, we measured the polysome/monosome ratios (P/M; estimate of translational efficiency) and ribosome content (P + M; estimate of translational capacity) for all polysome profiles of aal1-pOE, aal1Δ and control cells (Figs. 4A,B and EV4B). The P/M ratios did not show significant changes in aal1Δ or aal1-pOE cells (Fig. 4D). However, aal1 overexpression and deletion led to decreased and increased cellular ribosome contents, respectively, relative to the respective controls; however, these changes were only significant in proliferating and early stationary phase cells (Fig. 4E). We also quantified these parameters in the ectopic expression experiments in Fig. 4C and Fig. EV4C. Again, the differences were more substantial for ribosome content than for P/M ratios, with

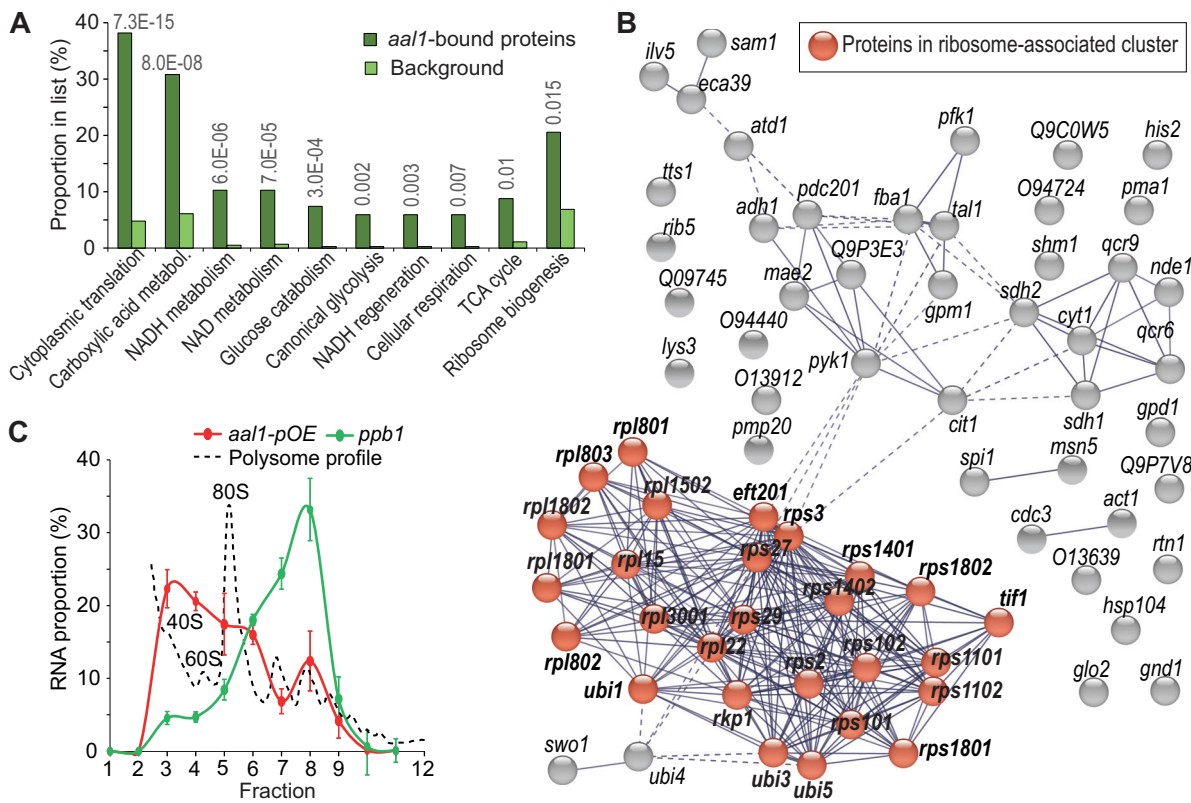

**Figure 3. *aal1* is associated with ribosomes.**

(A) Graph showing the proportions of genes with representative GO Biological Process terms among the 68 *aal1*-bound proteins compared to all *S. pombe* proteins (background). GO terms were selected for non-redundancy, specificity, and significance, with the respective enrichment FDRs shown above. FDRs were calculated with the Benjamini and Hochberg method in AnGeLi (Bitton et al, 2015). The gene list used is provided in Dataset EV2. (B) Analysis of protein–protein interaction network with STRING (Szklarczyk et al, 2021) for the 68 *aal1*-bound proteins reveals that these proteins are more connected with each other than expected by chance ($p = 1.0e{-}16$). Red balls: proteins associated with ribosomes. Dashed lines indicate lower confidence interactions. The following STRING parameters were used: active interaction sources = only experiments and databases; minimum required interaction score = high confidence (0.70); cluster with MCL; inflation = 3. (C) Polysome fractionation followed by RT-qPCR for *aal1-pOE* cells shows that *aal1* (red curve) mainly occurs with free ribosomal subunits (40S/60S) and monosomes (80S). The *ppb1* control mRNA (green curve) mainly occurs in polysomes. The corresponding polysome profile is shown as a black dashed line. Enrichment was calculated relative to the free RNA Fractions 1 and 2 (Bachand et al, 2006). The plot shows the mean ± SE of three independent repeats of *aal1* overexpressing cells during exponential growth in minimal medium. Source data are available online for this figure.

ectopic overexpression of *aal1* significantly reducing the ribosome content of *aal1Δ* cells (Fig. 4F). Together, these results suggest that *aal1* can act in trans to reduce the cellular ribosome content, while the absence of *aal1* increases the ribosome content.

Given the negative effects of *aal1* on ribosome content (Fig. 4A–F) and cell growth (Fig. 1D), we postulated that *aal1* inhibits protein translation. To directly test this possibility, we measured protein synthesis using puromycin incorporation assays. Puromycin is an aminoacyl-tRNA analogue that allows the labelling and detection of nascent peptide chains, which can then be quantified using an anti-puromycin antibody (Methods). In these assays, *aal1-pOE* cells showed reduced protein translation compared to control cells (Fig. 4G). Conversely, *aal1Δ* cells showed only a subtle increase in protein translation (Fig. 4G). The latter result is similar to the weak effect on cell growth in *aal1Δ* cells (Fig. 1E) and is consistent with *aal1* being hardly expressed in proliferating cells (Marguerat et al, 2012). Together, these findings indicate that *aal1* can reduce the cellular content of ribosomes and, thus, the capacity for protein translation.

## *aal1* binds to the *rpl1901* mRNA encoding a ribosomal protein whose repression prolongs lifespan

To further dissect how *aal1* might affect the cellular ribosome content, we used RNA-seq to examine its effects on the transcriptome. We analysed the differential gene expression between *aal1-pOE* and empty-vector control cells after 6 days in stationary phase, revealing 79 induced and 248 repressed transcripts. The overexpression of *aal1* led to repressing ribosome- and translation-related transcripts, including 127 of 135 mRNAs encoding *S. pombe* ribosomal proteins (Fig. 5A; Dataset EV3). The induced genes, on the other hand, were not enriched in any biological processes. Thus, *aal1* can directly or indirectly down-regulate the mRNAs for critical proteins involved in translation.

Some lncRNAs can regulate genes post-transcriptionally, e.g. through partial base-pairing between a lncRNA and target mRNAs to promote mRNA degradation (Gong and Maquat, 2011) or inhibit translation (Yoon et al, 2012). To test whether *aal1* binds to other RNAs, we used the pull-downs from the ChIRP-MS

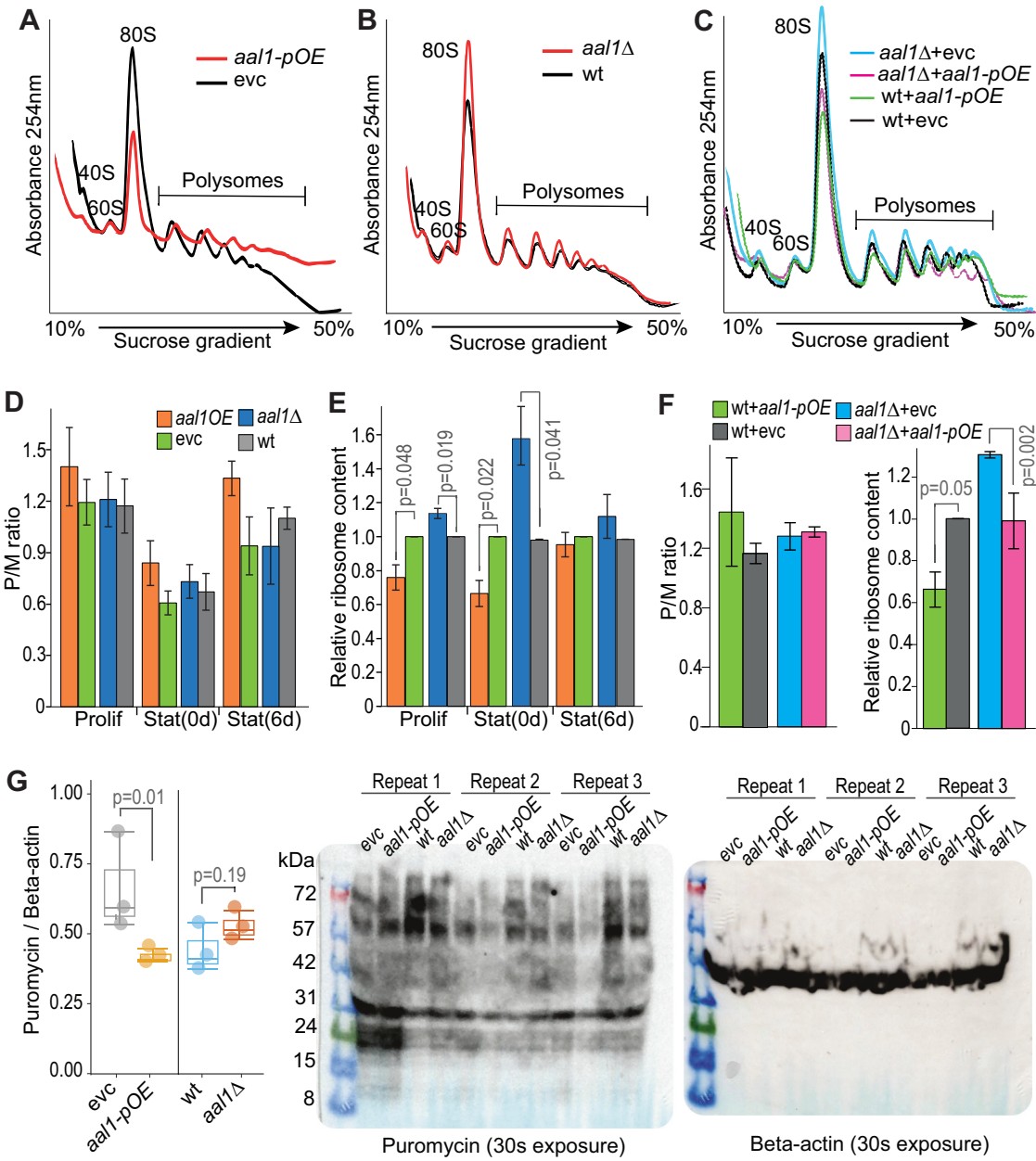

**Figure 4. _aal1_ reduces the ribosome content and translational capacity.**

(A) Polysome profiling for proliferating _aal1-pOE_ and empty-vector control (evc) cells in minimal medium. The two profiles are aligned at the lowest points of the monosome peaks, corresponding to the baseline. The polysome profiles from two independent biological repeats are shown in Fig. EV4B. (B) Polysome profiling for proliferating _aal1Δ_ and wild-type cells as in (C). The polysome profiles from two independent biological repeats are shown in Fig. EV4B. (C) Polysome profiling for proliferating cells in minimal medium. Ectopic expression of _aal1_ in _aal1Δ_ background (_aal1Δ + aal1-pOE_) reduces the increased ribosome content observed in _aal1Δ_ cells to a level similar to wild-type cells expressing an empty vector (wt+evc). The polysome profiles from two independent biological repeats are shown in Fig. EV4C. (D) Polysome/monosome (P/M) ratio in proliferating cells (Prolif), at the onset of stationary phase, Stat(0d), and after 6 days in stationary phase, Stat(6d), in _aal1-pOE_, empty-vector control (evc), _aal1Δ_ and wild type (wt) cells as indicated. Bars show mean ± SE of three independent repeats, with statistical significance determined using a two-sample t-test. (E) Relative ribosome content (Polysome+Monosome) in proliferating cells (Prolif), at the onset of stationary phase, Stat(0d), and after 6 days in stationary phase, Stat(6d), in _aal1-pOE_, relative to empty-vector control (evc) and _aal1Δ_ relative to wild type (wt) cells as indicated. Bars show mean ± SE of three independent repeats, with statistical significance determined using a two-sample t-test against the respective controls. The differences at Stat(6d) are not significant, likely reflecting that the ribosome content is generally low at this stage, which makes it harder to obtain reproducible measurements. (F) Polysome/monosome (P/M) ratios and ribosome content as in (D) and (E) for ectopic expression experiments shown in (C). Relative ribosome content determined relative to wt+evc. (G) Measurement of protein translation using puromycin labelling of proliferating cells. Left: graph showing anti-puromycin signal relative to anti-beta-actin signal quantified from chemiluminescence images for empty-vector control (evc) and _aal1-pOE_ cells as well as wild type (wt) and _aal1Δ_ cells as indicated. The quantified data were analysed with a linear regression model, which includes strain and batch as variables, and the significance was determined with a t-test. Data are shown as medians ± 95% confidence intervals. Right: Blots for signal quantification using anti-puromycin (left) and anti-beta-actin (right) antibodies for the 3 biological repeats of all strains as indicated above. Marker protein sizes in kilodaltons are indicated on the left. Source data are available online for this figure.

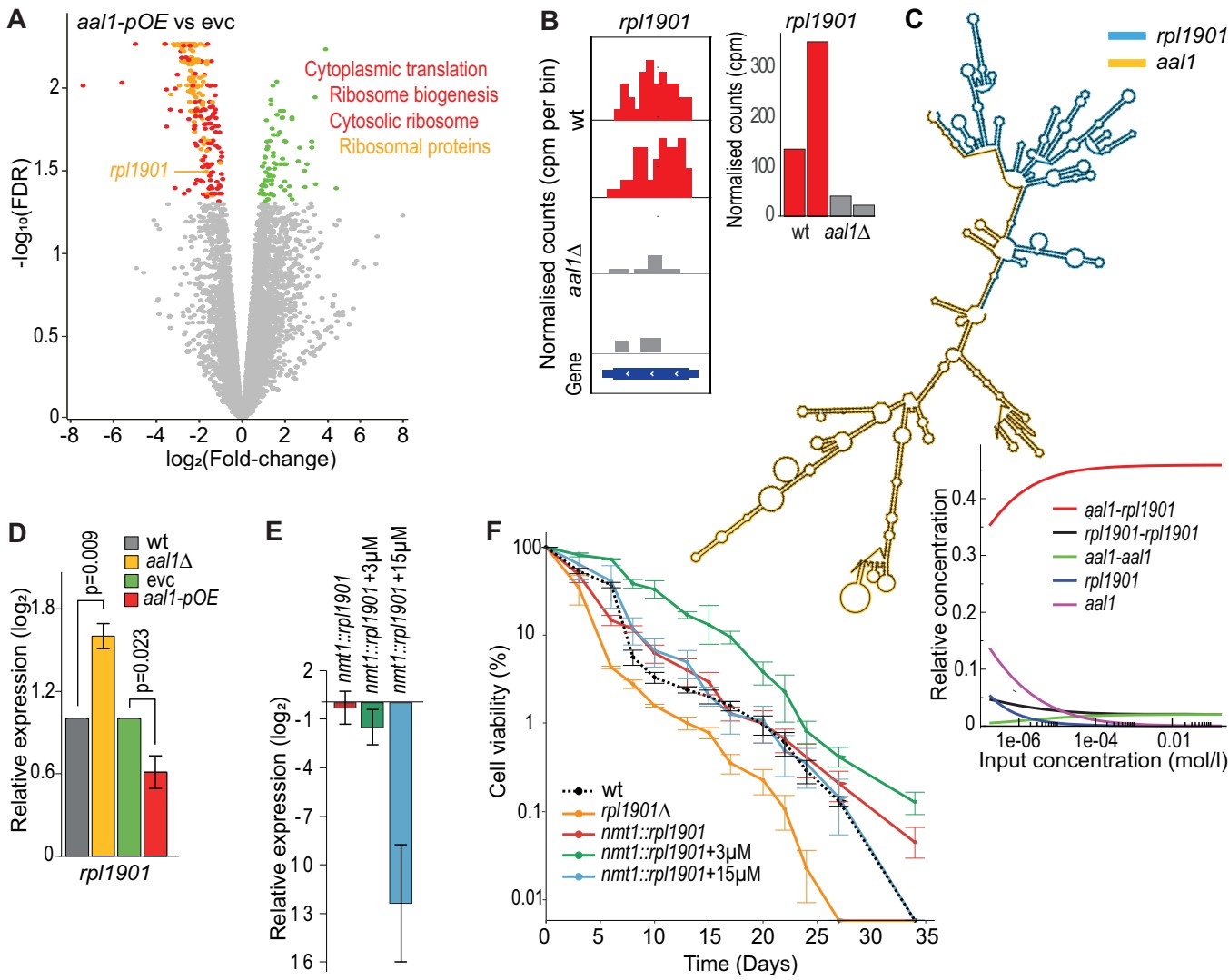

**Figure 5.** *aal1* binds to the *rpl1901* mRNA whose repression prolongs lifespan.

(A) Volcano plot of RNA-seq data comparing transcript levels in *aal1-pOE* relative to empty-vector control cells during stationary phase (Day 6) in minimal medium. The 248 repressed genes (red) are enriched in the GO terms 'Cytoplasmic translation' (139 genes; FDR: 5.7E−135), 'Ribosome biogenesis' (74 genes; FDR: 8.6E−32), and 'Cytosolic ribosome' (134 genes; FDR: 9.2E−178). The 127 repressed ribosomal protein genes are highlighted in orange with *rpl1901* indicated. The 79 induced genes (green) are not enriched in any GO terms. RNA-sequencing was performed with three biological replicates per genotype (Dataset EV3). (B) The ribosomal-protein mRNA *rpl1901* is enriched in two independent repeats of ChIRP pull-downs from wild-type (wt, red) compared to *aal1Δ* (grey) cells. Left panels: strand-specific ChIRP-seq reads in counts per million (cpm) for 50 bp bins across *rpl1901*, visualized using IGV tracks (Ge et al, 2020) and deepTools (Ramirez et al, 2016). Right panel: normalised read counts (cpm) for *rpl1901*. (C) In silico predictions of *aal1-rpl1901* interaction using the ViennaRNA package (Lorenz et al, 2011) with RNAcofold (Lorenz et al, 2016). Top: Predicted *aal1-rpl1901* heterodimer structure showing the interaction sites along with potential RNA secondary structures. Bottom: Concentration dependency plot of RNA-RNA interactions showing the computed homo- and hetero-interactions of RNAs for concentration relative to each other (y-axis) and different input concentrations (x-axis), with predicted equilibrium concentrations for the two monomers, *aal1* and *rpl1901*, the two homodimers, *aal1-aal1* and *rpl1901-rpl1901*, and the *aal1-rpl1901* heterodimer. (D) RT-qPCR experiment to quantify RNA levels of *rpl1901* in proliferating wild-type (wt), *aal1Δ*, empty-vector control (evc), and *aal1-pOE* cells. Expression was normalised to *act1* and shown relative to the expression in the respective controls. Bars show the mean ± SE, with the indicated *p*-values determined by t-test. (E) RT-qPCR experiment to quantify expression of *rpl1901* under the thiamine-repressible *P41nmt1* promoter at its native locus (*nmt1::rpl1901*), in the absence or presence of 3 or 15 μM thiamine as indicated. Expression was normalised to *act1* and shown relative to wild-type expression. Bars show the mean ± SE of three replicates. (F) Chronological lifespan assays for cells expressing *rpl1901* at different levels. Only the moderate repression of *rpl1901* with 3 μM thiamine promotes lifespan extension. Experiments were performed in minimal medium. Assays and analyses as in Fig. 1C. Source data are available online for this figure.

experiment to sequence any RNAs associated with *aal1* (ChIRP-seq) (Chu et al, 2012). As expected, *aal1* itself was the top-enriched RNA in these pull-downs compared to *aal1Δ* control cells, but other RNAs, both coding and non-coding, also appeared to be enriched (Fig. EV5A; Dataset EV4). Among these RNAs were four

encoding ribosomal proteins, of which *rpl1901* was most consistently enriched (Fig. 5B). To test the plausibility of an RNA-RNA interaction between *aal1* and *rpl1901*, we performed an in silico analysis using the ViennaRNA Package. This analysis predicted that *aal1* has a high potential to interact with *rpl1901* across four

regions, involving over 40 base pairs, and at very low concentrations (Fig. 5C). Among the other three mRNAs encoding ribosomal proteins, *rpl1802* showed predicted potential to interact with *aal1*, although less convincingly than for *rpl1901* (Fig. EV5B).

Like most other ribosomal-protein genes, *rpl1901* was repressed upon *aal1* overexpression in chronologically ageing cells (Fig. 5A). To quantitatively analyse how *aal1* affects *rpl1901* mRNA levels, we performed RT-qPCR assays in both *aal1Δ* and *aal1-pOE* cells during proliferation. RNA levels of *rpl1901* were modestly (~2-fold) but significantly repressed in *aal1-pOE* cells and induced in *aal1Δ* cells (Fig. 5D). This result is consistent with the possibility that *aal1* directly represses the *rpl1901* mRNA levels, which could contribute to the *aal1*-mediated reduction in ribosome content and lifespan extension.

We wondered whether repression of *rpl1901* might be involved in the *aal1*-mediated lifespan extension. We first generated a deletion mutant of *rpl1901* (*rpl1901Δ*) and showed that *rpl1901Δ* cells featured substantially shorter CLS and slower growth than wild-type cells (Fig. EV5C). However, *rpl1901* was completely absent in this deletion mutant, whereas *rpl1901* expression was only moderately repressed in response to *aal1* (Fig. 5D). Therefore, we tested whether a moderate repression of *rpl1901* might extend the lifespan. To this end, we expressed *rpl1901* under the thiamine-repressible *P41nmt1* promoter in its native locus (*nmt1::rpl1901*) to modulate *aal1* repression by supplementing various doses of thiamine. The expression levels of *nmt1::rpl1901* in the absence of thiamine were comparable to the *rpl1901* levels under its native promoter (Fig. 5E). Addition of 3 μM thiamine led to a ~ 2-fold reduction in *nmt1::rpl1901* levels similar to the repression of *rpl1901* observed in *aal1-pOE* cells, while 15 μM thiamine led to a much stronger repression (Fig. 5D,E). Expression of *nmt1::rpl1901* prolonged the lag period but not the growth rate, while in the presence of 3 or 15 μM thiamine, the growth rates were modestly but significantly reduced (Fig. EV5D). Notably, only the ~2-fold repression of *rpl1901* with 3 μM thiamine led to a substantial extension of the CLS, while the other conditions, leading to no or much stronger repression of *rpl1901*, showed similar or shorter CLS compared to wild-type cells (Fig. 5F). We conclude that only a moderate (~2-fold) repression of *rpl1901* but not a stronger repression is beneficial for cellular longevity, mirroring the *rpl1901* repression and lifespan extension seen in *aal1-pOE* cells.

### *aal1* can extend the lifespan of flies

Ribosomal proteins and most of the translational machinery are highly conserved across eukaryotes, and reducing protein translation to non-pathological levels prolongs lifespan in different model organisms (MacInnes, 2016; Rallis and Bähler, 2013; Solis et al, 2018; Turi et al, 2019). Although lncRNAs show little or no sequence conservation between species, their functional principles can be conserved (Herman et al, 2022; Ulitsky and Bartel, 2013). We wondered whether *aal1* might have the potential to extend lifespan in a multicellular eukaryote, as it does in *S. pombe* cells. We tested this possibility in the fruit fly, *Drosophila melanogaster*. We first ensured the expression of *aal1* in *Drosophila* by confirming the presence of the RNA of the expected size in female flies where *UAS-aal1* was driven by the ubiquitous, constitutive *GAL4* driver *daughterlessGAL4* (Fig. EV6A). We further observed that such ubiquitous expression of *aal1* in *Drosophila* throughout

development resulted in a significant reduction in the number of flies that reached adulthood compared to control flies, indicating that *aal1* expression negatively affects development (Fig. EV6B). To assess the impact of *aal1* on *Drosophila* ageing, we focused on the gut because *aal1* expression in yeast represses protein translation, and interventions that repress translation in the fly gut have pro-longevity effects (Filer et al, 2017; Martinez Corrales et al, 2020). To avoid any harmful effects on development, we used the mid-gut specific RU486 inducible driver (*TIGS*) in adult females, which can drive *aal1* expression upon feeding the inducer, RU486. Remarkably, this induction of *aal1* significantly extended the medium lifespan of female flies (Fig. 6A). The feeding of the inducer itself did not affect the lifespan in the *TIGS* driver-alone control (Fig. 6B), indicating that the extension is not an artefact of RU468 feeding. Although *aal1* is not conserved in *Drosophila*, it has a high predicted potential to dimerize with the *RpL19* mRNA, encoding the highly conserved *Drosophila* ortholog of Rpl1901 (Fig. EV6C). These findings show that *aal1* can promote longevity in *Drosophila* and raise the possibility that the functional principle of *aal1* is conserved and that *aal1* lncRNA counterparts might exist in multicellular eukaryotes.

## Discussion

This study presents the functional characterization of a lncRNA, *aal1*, which can lower the ribosomal content and extend the lifespan of non-dividing fission yeast cells. The genetic and protein interactions of *aal1*, along with its various phenotypes, point to an involvement in protein translation. The *aal1* RNA localizes to the cytoplasm, and its ectopic expression from a plasmid can rescue *aal1* deletion phenotypes and trigger key phenotypes like decreased ribosome content and protein translation, reduced growth, and increased lifespan, indicating that *aal1* functions in trans. While further work is required to dissect the specific mechanisms of *aal1* function, we propose a simple model for how *aal1* might exert its roles in protein translation and longevity (Fig. 6C). Below, we discuss the key steps of this model in the context of our results and published findings.

The *aal1* RNA is induced in non-dividing cells, possibly by inhibiting its degradation via the RNAi and nuclear-exosome pathways that repress *aal1* in proliferating cells (Atkinson et al, 2018) (Fig. 1A,B). Then, *aal1* binds to other RNAs, including the *rpl1901* mRNA that encodes a ribosomal protein (Fig. 5B,C). This interaction may lead to a ~2-fold reduction of *rpl1901* mRNA levels (Fig. 5D), and such a modest reduction is critical and sufficient to extend the CLS (Fig. 5F). Overexpression of *aal1* leads to a similar *rpl1901* reduction (Fig. 5D) and CLS extension (Fig. 1C).

How might *aal1* repress the *rpl1901* mRNA levels? It is known that lncRNAs can regulate genes post-transcriptionally through base-pairing with target mRNAs to promote their degradation or inhibit their translation (Gong and Maquat, 2011; Yoon et al, 2012). The *aal1* RNA is associated with ribosomes (Fig. 3), as also reflected in the genetic interactions (Fig. 2A), which may point to its mode of action. Several RNAs interact with ribosomes in different organisms (Carlevaro-Fita et al, 2016; Duncan and Mata, 2014; Guttman et al, 2013; Wang et al, 2017), reflecting their translation into peptides (Duncan and Mata, 2014; Wang et al, 2017), degradation

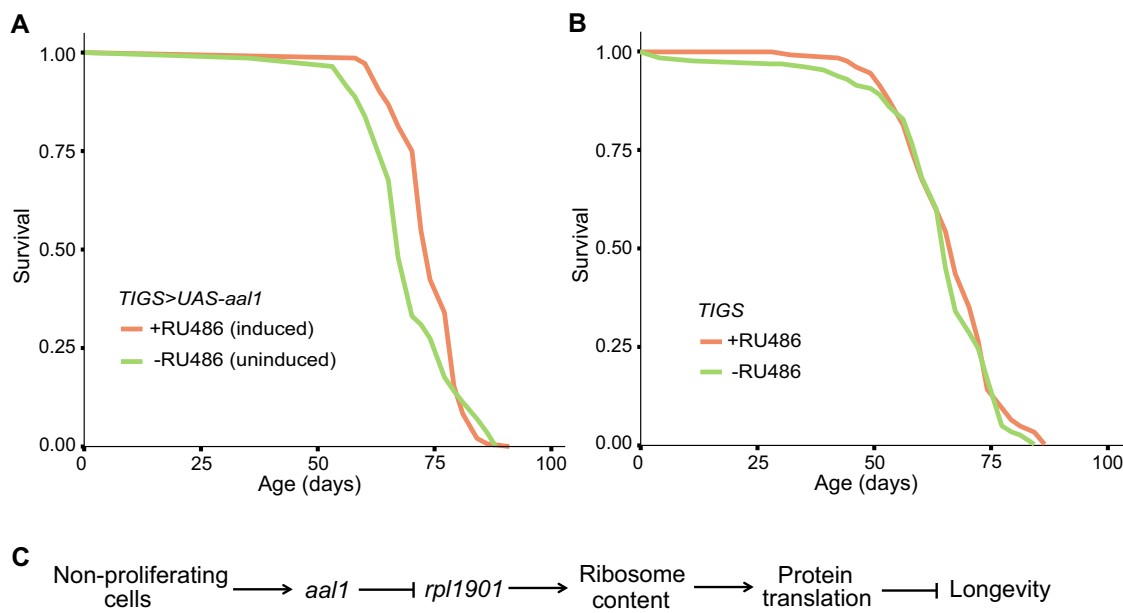

**Figure 6. aal1 expression prolongs the lifespan in flies.**

(A) Lifespans of female *Drosophila* with *UAS-aal1* induced in adult midguts with the gut-specific *TIGS* driver by RU486 feeding to activate expression (−RU486: $n = 142$ dead/8 censored flies, +RU486: $n = 144$ dead/6 censored flies, $p = 0.003$, log-rank test). (B) Lifespans of control females carrying the *TIGS* driver alone with or without RU486 feeding (−RU486: $n = 131$ dead/19 censored flies, +RU486: $n = 144$ dead/6 censored flies, $p = 0.3$, log-rank test). (C) Simple model for *aal1* function in lifespan extension via inhibition of *rpl1901* expression and protein translation (details in Discussion). Source data are available online for this figure.

via nonsense-mediated decay (NMD) (Quinn and Chang, 2016), or involvement in translational control (Essers et al, 2015; Guttman et al, 2013; Yoon et al, 2012). Available data indicate that *aal1* is not detectably translated in cells undergoing meiosis (Duncan and Mata, 2014), when *aal1* is induced (Fig. 1A), and *aal1* is not induced in mutants of the NMD-pathway gene *upf1* (Atkinson et al, 2018). It is possible, however, that the interaction of *aal1* with the *rpl1901* mRNA during its translation leads to NMD-mediated degradation of *rpl1901*. We find that *aal1* is more associated with monosomes than polysomes (Fig. 3C), which might suggest a role before translational elongation, although older cells also showed a substantial proportion of *aal1* in polysomes (Fig. EV4A). However, monosomes can also reflect the active translation of short open reading frames (<590 nt), where translational initiation takes relatively more time than translational elongation (Biever et al, 2020; Heyer and Moore, 2016). The *rpl1901* mRNA is 582 nt long, so our results are consistent with the possibility that *aal1* associates with *rpl1901* in the context of translating ribosomes. Conversely, RNAi does not seem to be directly involved in this repression, because the *rpl1901* mRNA, unlike *aal1* (Fig. 1A), is reduced in RNAi mutants (Atkinson et al, 2018). However, it is of course possible that other or additional pathways may contribute to an *aal1*-mediated *rpl1901* mRNA repression, including transcriptional control.

Compared to *rpl1901*, *aal1* is more lowly expressed, raising a potential paradox of regulatory stoichiometry. But in non-dividing cells, *rpl1901* is expressed at only ~6 molecules per cell (Marguerat et al, 2012), which may be similar to *aal1* that is induced in this condition (Fig. 1A). Furthermore, several lncRNAs, including Xist and NORAD, form biomolecular condensates to facilitate their

substoichiometric action, as these phase-separated condensates can concentrate factors (Somasundaram et al, 2022; Unfried and Ulitsky, 2022). Indeed, the low levels of the Xist RNA relative to its abundant targets are essential to maintain target specificity, suggesting that low levels are critical for lncRNA function (Jachowicz et al, 2022; Markaki et al, 2021). Examples of ribonucleoprotein condensates are stress granules, processing bodies, and the nucleolus (Hirose et al, 2023). For example, in *S. pombe*, *meiRNA*, a lncRNA expressed at only 0.21 molecules per cell (Marguerat et al, 2012), forms a nuclear body with the RNA-binding protein Mei2 to promote meiotic differentiation (Shichino et al, 2014). Recently discovered translation factories are cytoplasmic bodies accumulating many copies of certain RNAs (Basyuk et al, 2021). Notably, we observed that *aal1* localizes in cytoplasmic foci (Fig. 2B), which might reflect ribonucleoprotein bodies facilitating molecular interactions between *aal1*, *rpl1901*, and ribosomes to trigger NMD-mediated degradation of *rpl1901*.

We propose that the *aal1*-mediated repression of *rpl1901* leads to the global repression of other ribosomal protein genes, which in turn reduces the cellular ribosome content and protein translation, thus promoting longevity (Fig. 6C). Gene-specific feedback mechanisms can increase the expression of individual ribosomal proteins present at substoichiometric amounts, but these mainly function through the regulation of splicing and/or mRNA stability rather than transcription or translation (Petibon et al, 2021). The *rpl1901* gene does not have introns, and *aal1*-mediated degradation might attenuate protein translation. This could be achieved by global negative feedback regulation of other ribosomal protein genes in response to lowered *rpl1901* levels (Kong and Lasko, 2012; Warner, 1999). Consistent with this possibility, a ~2-fold reduction

in *rpl1901* levels is sufficient to reduce cell growth (Fig. EV5D). Moreover, *aal1* overexpression leads to the repression of 127 of 135 mRNAs encoding the proteins of the *S. pombe* cytoplasmic ribosome (Fig. 5A) and triggers a reduction in cellular ribosome content and protein translation (Fig. 4) as well as growth (Fig. 1D). Ribosome content, translational capacity, and growth rate are linked in yeast and other cells, e.g. via TORC1 signalling (Bjorkeroth et al, 2020; Ingolia et al, 2009; Liu and Sabatini, 2020; Martinez-Miguel et al, 2021; Warner, 1999). As would be expected for the proposed mechanism, however, the *aal1*-mediated growth reduction is independent of TORC1 signalling (Fig. EV2). Attenuation of protein translation to non-pathological levels is known to prolong lifespan and delay ageing-associated diseases in model organisms, although the detailed nature of these relationships is still unclear (Kaeberlein and Kennedy, 2011; MacInnes, 2016; Pan et al, 2007; Rallis and Bähler, 2013; Solis et al, 2018; Steffen and Dillin, 2016; Turi et al, 2019).

It is possible that *aal1* functions via additional or other processes than those proposed in Fig. 6C. However, it is notable that modulation of *rpl1901* expression is sufficient to mirror the key phenotypes of *aal1*. Intriguingly, while mutants in most ribosomal proteins do not appear to affect ageing (Polymenis, 2020), orthologs of Rpl1901 (RPL19) are among only four ribosomal proteins shown to affect ageing in budding yeast and worms where the loss of RPL19 function extends lifespan (Curran and Ruvkun, 2007; Hansen et al, 2007; Kaeberlein et al, 2005). RPL19 is also among the few ribosomal protein genes that are up-regulated in embryonic stem cells of human blastocysts and hepatocellular carcinoma (Adjaye et al, 2005; Rao et al, 2021), and post-transcriptional silencing of RPL19 alleviates human prostate cancer by controlling the translation of a subset of transcripts (Bee et al, 2011). Ribosomal proteins may affect ageing in multiple ways, e.g. by altering global translation rates, functioning in specialized ribosomes, or influencing the protein biogenesis machinery (Janssens et al, 2015; Maitra et al, 2020; Mittal et al, 2017; Steffen et al, 2008).

Protein translation is highly energy-demanding and tightly regulated by conserved mechanisms to adjust supply with demand in varying environmental or physiological conditions (MacInnes, 2016). Non-coding RNAs, including the well-known rRNAs, tRNAs, snoRNAs and miRNAs, function in protein translation and its regulation. More recently, lncRNAs have emerged as regulators of ribosome biogenesis and translation. For example, SLERT controls nucleolar phase separation and rRNA expression (Wu et al, 2021), LoNA inhibits translation and rRNA expression at multiple levels (Li et al, 2018), while other lncRNAs promote rDNA chromatin compaction to repress rRNA transcription in quiescent cells (Bierhoff et al, 2014). Several ribosome-associated non-coding RNAs are emerging as translational regulators in response to stress (Carvalho Barbosa et al, 2020; Pecoraro et al, 2022). The nematode *daf-2* mutants in an insulin receptor-like gene display similar phenotypes to *aal1*, including slow growth and extended lifespan (Kenyon et al, 1993), along with low ribosomal-protein levels and reduced protein synthesis (Stout et al, 2013; Walther et al, 2015). Interestingly, the *tts-1* RNA is required for the lifespan extension in *daf-2* mutants and is mainly associated with the monosomal fraction of ribosomes (Essers et al, 2015), akin to *aal1*, raising the possibility that these lncRNAs play similar mechanistic roles.

To our surprise, the expression of *S. pombe aal1* in *Drosophila* leads to a significant lifespan extension in *Drosophila* (Fig. 6A). Interestingly, both the organ whence the longevity effects arise, namely the gut, and the magnitude of the effects are comparable between *aal1* expression and the partial inhibition of RNA polymerase I, which transcribes the pre-rRNA (Martinez Corrales et al, 2020). Thus, although *aal1* is not conserved in *Drosophila*, this RNA can substantially promote the longevity of both single yeast cells and multicellular flies. Furthermore, given that ribosomal proteins and translational processes are highly conserved from yeast to humans, it is plausible that *aal1* functions through related mechanisms in both yeast and flies and perhaps can take on the role of a native fly lncRNA, which functions within a conserved regulatory framework. Indeed, in silico analysis predicts that *aal1* has a high potential to dimerize with the *Drosophila RpL19* mRNA, encoding the highly conserved ortholog of *S. pombe* Rpl1901 (Fig. EV6C). Structure-function constraints for lncRNAs are more relaxed than for proteins (Quinn and Chang, 2016), and RNAs with different sequences may interact with similar proteins. It would be interesting to test whether *aal1* binds the *RpL19* mRNA in *Drosophila* to inhibit the cellular ribosome content. The finding in *Drosophila* raises the intriguing possibility that other organisms use analogous lncRNAs and related mechanisms to control the translational capacity of certain cells, and such lncRNAs would potentially be effective targets for future RNA-based therapies to delay ageing and associated diseases.

## Methods

### Fission yeast strains and culture conditions

The standard *S. pombe* lab strain *972 h⁻* containing *leu1-32* was used to transform strains, for experiments using *aal1-pOE* and empty-vector control. Two independent deletion strains *aal1Δ1* and *aal1Δ2* were generated by CRISPR-Cas9 using distinct sgRNAs (Rodriguez-Lopez et al, 2016) in the *972 h⁻* strain. Strains *aal1::natMX6* (*aal1Δ*, used for SGA), *kanMX6:P41nmt1-aal1* (*aal1-gOE*, used for smFISH), *natMX6:P41nmt1-aal1* (*aal1-gOE*, used for SGA), *rpl1901Δ::kanMX6* (*rpl1901Δ*) and *natMX6:P41nmt1-rpl1901* (*nmt1::rpl1901*, used in Fig. 5E,F) were constructed as described (Bähler et al, 1998) in the *972 h⁻* background and deletion/overexpression primers were designed using the Pombe PCR Primer Programs at http://www.bahlerlab.info/resources/. For the *rpl1901* deletion mutant, the PCR reaction to create the insertion cassettes included 8% DMSO (Invitrogen). The *aal1-pOE* (*P41nmt1-aal1*) and *SPNCRNA.401-pOE* (*P41nmt1-SPNCRNA.401-pOE*) ectopic over-expression strains, driven by the medium-strength *nmt1* promoter (Bähler et al, 1998), were generated by PCR amplification of the predicted full-length *SPNCRNA.1530* and *SPNCRNA.401* sequences, respectively, as annotated in PomBase (Lock et al, 2019), using the high-fidelity Phusion DNA polymerase (NEB) and cloning into the pJR1-41XL vector (Moreno et al, 2000) with the CloneEZ PCR Cloning Kit (GenScript). Plasmids were checked by PCR for correct insert size and also by Sanger sequencing (Eurofins Genomics) using a universal forward primer (5′ CGGATAATG-GACCTGTTAATCG 3′) for the pJR1-41XL plasmid upstream of

**Reagents and tools table**

| Reagent/Resource | Reference or Source | Identifier or Catalog Number |
|---|---|---|
| **Experimental Models** | | |
| 972h- (*S. pombe* wild type) | Bähler lab | |
| 972h-, *leu1-32* (*S. pombe*) | Bähler lab | |
| *P41nmt1-aal1* (*S. pombe*, ectopic) | This study | |
| CRISPR-Cas9 *aal1Δ* (*S. pombe*) | This study | |
| *aal1Δ::natMX6* (*S. pombe*) | This study | |
| *kanMX6:P41nmt1-aal1* (*S. pombe*) | This study | |
| *natMX6:P41nmt1-aal1* (*S. pombe*) | This study | |
| *rpl1901Δ::kanMX6* (*S. pombe*) | This study | |
| *natMX6:P41nmt1-rpl1901* (*S. pombe*) | This study | |
| *P41nmt1-SPNCRNA.401* (*S. pombe*, ectopic) | This study | |
| White Dahomey (*D. melanogaster*) | Alic lab | |
| GAL4-TIGS (*D. melanogaster*) | Alic lab | |
| UAS-aal1 (*D. melanogaster*) | This study | |
| **Recombinant DNA** | | |
| *P41nmt1-aal1* | This study | |
| *P41nmt1-SPNCRNA.401* | This study | |
| UAS-aal1 | This study | |
| pJR1-41XL | Moreno et al, 2000 | |
| **Antibodies** | | |
| Anti-puromycin | Millipore | 12D10 |
| Anti-beta-actin | Abcam | Ab8224 |
| **Oligonucleotides and other sequence-based reagents** | | |
| Universal forward primer for pJR1-41XL plasmid upstream of the cloning site | This study | 5'CGGATAATGGACCTGTTAATCG 3' |
| *aal1*_qRT_Forward | This study | CGCGTATTCCCTTTGGTGTT |
| *aal1*_qRT_Reverse | This study | GGGTCCTACATTCTTGAGGCT |
| *ckb2*_qRT_Forward | This study | TCCTAAGCGCTCAATCCCTTG |
| *ckb2*_qRT_Reverse | This study | CCATTGCATTCGAGGTCCTG |
| *ctu1*_qRT_Forward | This study | GACTACATGTGAGCGTTGCG |
| *ctu1*_qRT_Reverse | This study | CACTTCCCAAACCGAGACCA |
| *act1*_qRT_Forward | This study | TCCTCATGCTATCATGCGTCTT |
| *act1*_qRT_Reverse | This study | CCACGCTCCATGAGAATCTTC |
| *ppb1*_qRT_Forward | This study | CACCTGTCACTGTATGCGG |
| *ppb1*_qRT_Reverse | This study | GATCCACGTAATCTCCAAGGAA |
| *rpl1901*_qRT_Forward | This study | GACTTGCTGCATCCGTCC |
| *rpl1901*_qRT_Reverse | This study | TAACCAAACCGTCCTTAATCAAC |
| *aal1*-Gene_specific_RT-reaction_primer (reverse) | This study | CATGGAATACACTATCACGACAACAT |
| *ppb1*-Gene_specific_RT-reaction_primer (reverse) | This study | TCAGTCATTCTACTCGCGTAA |
| *aal1_D.mel*-cDNA-PCR_Forward | This study | TGAAACATTGTCATCTCTTGC |
| *aal1_D.mel*-cDNA-PCR_Reverse | This study | GGAATACACTATCACGACAAC |
| Control_Reverse1_ Dm-cDNA-PCR | This study | CCGGAGTATAAATAGAGGCGC |
| Control_Reverse2_ Dm-cDNA-PCR | This study | CGGTACCCGCCCGGGATCAG |

| Reagent/Resource | Reference or Source | Identifier or Catalog Number |
|---|---|---|
| **Chemicals, Enzymes and other reagents** | | |
| EMM | Formedium | PMD0210 |
| YE medium | Formedium | PCM0110 |
| Rapamycin | Cayman Chemicals | 13346 |
| Caffeine | Sigma | C0750 |
| Cycloheximide | Sigma | C7698 |
| 0.1M PMSF | Sigma | 93482 |
| cOmplete Mini EDTA-free protease inhibitor cocktail | Roche | 11836170001 |
| Superase-In RNase inhibitor | Invitrogen | AM2696 |
| Dynabeads (magnetic, Streptavidin) | Invitrogen | 65001 |
| Biotin | Pierce | 29129 |
| Phusion DNA polymerase | NEB | M0530L |
| Turbo DNase | Invitrogen | AM2239 |
| DNaseI | Qiagen | 79254 |
| Trizol reagent | Invitrogen | 15596018 |
| QIAzol | Qiagen | 79306 |
| Puromycin | Gibco | A11139-02 |
| NuPAGE™ Bis-Tris Mini Protein Gels, 4–12% | Invitrogen | NP0322BOX |
| RU486 | Sigma | M8046 |
| 48-well microtiter plates (FlowerPlates) | m2p-labs | MTP-48-B |
| **Software** | | |
| R | R Core Team, 2020 | Ver. 4.0.3 |
| gitter (R-package) | Wagih and Parts, 2014 | Ver 1.1.3 |
| grofit (R-package) | Kahm et al, 2010 | Ver 1.1.1-1 |
| multcomp | Hothorn et al, 2008 | multcomp_1.4-23 |
| FastQC | https://www.bioinformatics.babraham.ac.uk/projects/fastqc/ | ver 0.11.2 |
| je | Girardot et al, 2016 | Ver. 1.2 http://gbcs.embl.de/je |
| STAR | Dobin et al, 2013 | ver. 2.6.1a, https://github.com/alexdobin/STAR |
| htseq | Anders et al, 2015 | ver 0.12.3, https://htseq.readthedocs.io/en/master/index.html |
| ImageQuant™ TL 10.2 analysis software | Amersham | ver. 10.2 |
| ImageJ | Schneider et al, 2012 | ImageJ 1.53k, Java 1.8.0_172 (64-bit) |
| edgeR (R-package) | Robinson et al, 2010 | Ver 3.32.1 & 3.46.0 |
| limma (R-package) | Ritchie et al, 2015 | limma_3.46.0 |
| Hisat2 | Kim et al, 2019 | Ver 2.2.1 |
| samtools | Danecek et al, 2021 | Ver 1.14 |
| Integrated Genome Viewer (IGV) | Thorvaldsdottir et al, 2013 | ver 2.8.0, https://software.broadinstitute.org/software/igv/ |
| deepTools | Ramirez et al, 2016 | ver 3.5.1, https://deeptools.readthedocs.io/en/develop/ |
| ViennaRNA Package | Lorenz et al, 2011 | ver. 2.5.0, https://www.tbi.univie.ac.at/RNA/documentation.html |
| RNAcofold | Lorenz et al, 2016 | Ver 2.6.4 |
| **Other** | | |
| Bead beater | MP Biomedicals | FastPrep24 |
| Sonicator | Diagenode | Bioruptor Pico |

| Reagent/Resource | Reference or Source | Identifier or Catalog Number |
|---|---|---|
| Colony Pinning robot | Singer Instruments | RoToR HDA |
| Semidry transfer system | BioRad | Trans-Blot Turbo |
| ImageQuant Imager | Amersham | ImageQuant800 |
| Sucrose Gradient Maker | Biocomp | Gradient Master |
| Gradient Fractionator | Teldyne ISCO | discontinued |
| Real-Time PCR System | Applied Biosystems | QuantStudio 6 Flex |
| Bioanalyzer | Agilent | Agilent 2100 |
| HiSeq2500 sequencing system | Illumina | Illumina HiSeq2500 sequencer |
| MiSeq sequencing system | Illumina | Illumina MiSeq sequencer |
| CloneEZ PCR Cloning Kit | GenScript | L00339 |
| Qiagen RNeasy Mini Ki | Qiagen | Current equivalent product 74634 |
| NEXTflex™ Rapid Directional qRNA-Seq™ Kit | PerkinElmer, formerly BiooScientific | 5130-11 |
| *S. pombe* Bioneer haploid deletion library | Bioneer | Ver 5.0 |
| *PomBase (S. pombe Genome database)* | Lock et al, 2019 | www.pombase.org |
| Department of Genetics Fly Facility | University of Cambridge | https://www.flyfacility.gen.cam.ac.uk/Services/Microinjectionservice/ |

the cloning site. Plasmids were transformed into *S. pombe* cells (*leu1-32 h-*), and leucine prototroph transformants were selected on solid Edinburgh Minimal Medium (EMM2, with 5 g/l $NH_4Cl$ as nitrogen source). An empty-vector control (evc) strain was created analogously. For *aal1Δ* rescue experiments, the *aal1Δ1* strain was recreated in a *leu1-32 h-* strain and either the *aal1-pOE* or evc plasmid was transformed. All strains were revived in rich YES medium (yeast extract with supplements and 3% glucose) from glycerol stocks. All experiments that included strains expressing genes under the transcriptional modulation of the *P41nmt1* promoter were grown in EMM2 or EMM supplemented with 3.75 g/l glutamate (EMMG). All cultures were grown at 32 °C with 170 rpm shaking in an INFORS HT Ecotron incubator unless indicated otherwise.

## Quantitative yeast growth assays in liquid culture

The assays were performed in a BioLector microbioreactor (m2p-labs) at 32 °C with 1000 rpm shaking and 85% humidity in 48-well microtiter plates (FlowerPlates). The starter cultures were grown to mid-exponential phase, diluted in EMMG to achieve initial $OD_{600}$ of ~0.05 and 1400 µl cultures were incubated in hexaplicates per treatment/genotype with well placements completely randomised to minimise any positional and border effects. For repression of the *P41nmt1* promoter 15 µM thiamine (Sigma) was used. Rapamycin (Cayman Chemical) alone experiments were performed with 300 ng/ml added after 8 h of initial growth, whereas combined drug experiments used 100 ng/ml rapamycin and 10 mM caffeine (Sigma) added at the start of incubation. Growth (biomass accumulation) was monitored in real time, measuring every 10 min until the cultures reached stationary phase. The growth data were normalised to time 0. Mean growth curves were fitted and growth rates and lag periods were calculated with *grofit* (Kahm et al, 2010). For the *aal1-pOE* vs evc +/− thiamine assays, statistical significance was determined with a one-way ANOVA

followed by Tukey's Honest Significant Difference test for all pairwise comparisons, using *R* (R Core Team, 2020) (ver 4.0.3) (45). Statistical significance of *aal1Δ* against wild-type comparisons were determined with a one-way ANOVA followed by Dunnett's test (*multcomp*) (Hothorn et al, 2008).

## Chronological lifespan assays

To determine the CLS, strains driven by a *P41nmt1* (*aal1-pOE*) promoter and respective controls (evc) were grown with or without 15 µM thiamine in EMMG medium, whereas deletion mutants (*aal1Δ*) and controls (wt) were grown in YES medium, each strain as three independent biological replicates. Chronological lifespan assays for the panels in Fig. 1C were performed as described (Roux et al, 2009). Day 0 was the day the cultures reached a stable maximal cell density. The percentages of viable cells were measured by serial dilution and plating in duplicate YES plates for each dilution. Colonies were counted, and the probable number of CFUs per ml was calculated. The percentage viability was calculated relative to the CFUs at Day 0 (100% cell survival). CFU measurements were made daily for YES cultures and every 3–4 days for EMMG cultures until cultures reached 0.1–1% of the initial cell survival. The means of three independent biological replicates, with each culture measured twice at each timepoint, are shown in the plots (error bars represent the standard error).

For CLS assays with cells expressing *rpl1901* at different levels (Fig. 4F), *rpl1901* was expressed under the thiamine-repressible *P41nmt1* promoter at its native locus (*nmt1::rpl1901*), in the absence or presence of 3 or 15 µM thiamine in EMMG medium. The percentages of viable cells were measured using a robotics-based colony-forming units (CFU) assay (Romila et al, 2021). A Poisson distribution-based model was used to obtain the maximum likelihood estimates for the number of CFUs, and percentage viability was calculated relative to that of the CFUs at Day 0 (100% cell survival).

## Polysome fractionation and RT-qPCR

Cells were grown in EMMG at 32 °C and 100 ml cultures were sampled for each timepoint. To block translation and capture translating ribosomes, cycloheximide (Sigma) was added to a final concentration of 100 µg/ml and incubated for 5 min with shaking. Cells were collected by centrifugation and lysed in lysis buffer (20 mM Tris-HCl pH 7.5, 50 mM KCl, 10 mM MgCl$_2$) supplemented with 100 µM cycloheximide, 1 mM DTT (Sigma), 20 U/ml SuperaseIn (Invitrogen) and protease inhibitors (Complete, EDTA-free, Roche). Lysis was performed with 0.5 mm acid-washed beads in a FastPrep instrument (MP, FastPrep24, Settings: speed, 6.0 m/s; adaptor, Quick Prep; time 20 s; 5 cycles with ≥5 min incubations on ice in between). The number of lysis cycles was increased to ~12 for stationary phase and ageing cells to achieve >80% lysis. The lysates were centrifuged at 17,000 × *g* for 5 min followed by another 15 min at 4 °C to remove cell debris, and the lysates were quantified in a Nanodrop (OD$_{260}$). Equal amounts of each lysate were loaded for the polysome fractionation. Then, 10–50% linear sucrose gradients were prepared with a Gradient Master (Biocomp) using 10% and 50% sucrose (Sigma) solutions prepared in lysis buffer freshly supplemented with 100 µM cycloheximide and 1 mM DTT. The lysates were carefully laid on top of the gradients and centrifuged in a SW-41Ti rotor in a Beckman L-80 ultracentrifuge at 35 K for 2 h 40 min at 4 °C. The tubes were processed in a Gradient Fractionator (Teldyne ISCO) with 55% sucrose as the chase solution. The polysome fractionation profiles were recorded and the fractions collected (~800–900 µl per fraction; 12–13 fractions per gradient) and immediately placed on ice. The area under the curve (AUC) of the monosome peak and the polysome peaks were measured with ImageJ (Schneider et al, 2012) for calculating the polysome:monosome ratios (P/M) and total ribosome content (P + M). The P/M was calculated by dividing the sum of the AUC of all polysomes divided by the AUC of the monosome, whereas P + M was the sum of both of these.

RNA was extracted with TRIzol reagent (Invitrogen) as per the manufacturer's recommendations. RNA was precipitated with isopropanol overnight at −20 °C. DNase digestion and subsequent RT-qPCR analysis were performed with 1 µg RNA as described below. The calculation of *aal1* and *ppb1* enrichment across polysome profiles was calculated as described (Bachand et al, 2006). Briefly, the threshold cycle (C$_T$) of each fraction was subtracted from the C$_T$ of maximum value (always either fraction 1 or 2) for each primer set. The resulting difference in threshold cycles (ΔC$_T$) was used to calculate the relative change in mRNA levels between fractions by calculating the $2^{\Delta CT}$ value. The mRNA distribution across the entire polysome profile was graphically presented as the percentage of mRNA in each fraction divided by the total amount of mRNA (sum of 12 fractions).

## RT-qPCR analysis

RNA was extracted using the TRIzol reagent (Invitrogen) as per manufacturer's recommendations. In-tube DNaseI (Turbo DNase, Invitrogen) digestion and subsequent reverse transcription (RT) was performed with 1 µg RNA. Random primed cDNA was prepared with SuperScript III reverse transcriptase (Invitrogen) as per standard protocols and all samples had an equivalent RT-reaction. Strand-specific RT with gene-specific primers were used for *aal1* transcript quantification with 120 ng/µl Actinomycin D added additionally in the RT reaction. RT-qPCR was performed in a QuantStudio 6 Flex Real-Time PCR System (Applied Biosystems) with Fast SYBR Green Master mix (Applied Biosystems), 1/5 diluted cDNA template and 250 nM primers as per manufacturer's recommendations. Samples were run in triplicates along with non-template and RT- controls and relative starting quantity was estimated using the ΔΔCt (Livak and Schmittgen, 2001) method. The *aal1* transcript levels across samples were normalized to the *ppb1* expression levels; *ppb1* is the least variable gene under many perturbations including stationary-phase and is comparatively lowly expressed (Pancaldi et al, 2010). All other protein-coding transcripts were normalised to *act1* expression levels. Melt curve analysis was performed following amplification to confirm the specificity of amplicons over primer dimers. All primer pairs were initially assessed in a standard curve for efficiencies, and primer pairs with efficiencies of 90–110% were used for RT-qPCR. All primers used are listed in Appendix Table S1.

## Genome-wide genetic interaction analyses (SGA)

The SGAs were performed as described (Rallis et al, 2017) using the *S. pombe* Bioneer haploid deletion library v5.0 consisting of 3420 deletion mutants. The *aal1Δ* (*aal1::natMX6 h⁻*) and *aal1-gOE* (*natMX6::nmt1:aal1 h⁺*) query strains with a nourseothricin resistance cassette (*nat*) were generated as described (Bähler et al, 1998). The *ade6Δ* strain (*ade6::natMX6 h⁻*), which does not alter the fitness of the mutant library under optimal conditions, was used as a control query strain serving as the equivalent of the fitness of the single mutants (Rallis et al, 2014). We assessed all pairwise gene interactions using colony size (growth rate) as a measure of double-mutant fitness relative to that of the *ade6Δ* control double mutants. The deletion library was revived in YES-agar, grown for 3 days at 32 °C and copied onto YES-agar with G418 (100 µg/ml). All query strains, including the control, were prepared in 384-well format in YES-agar with nourseothricin (100 µg/ml). The query strains were mated with the library mutant strains to create double mutants in 384-well format with a RoToR HDA pinning robot (Singer Instruments) in EMM-N-agar supplemented with adenine, uracil and leucine (100 mg/ml each). The plates were incubated at 25 °C for 3 days to allow mating and sporulation, followed by a 42 °C incubation for 3 days to kill the parental cells. Then, the colonies were copied onto YES-agar and incubated for 1 day to allow spores to germinate. When the colonies were sufficiently grown, they were pinned onto double selection plates: YES-agar with 100 µg/ml G418 and nourseothricin for *aal1Δ* SGA and EMMG-agar with 500 µg/ml G418 and nourseothricin for *aal1-gOE* SGA. The plates were incubated at 32 °C for 2 days and imaged using a EPSON V800 scanner as .jpg files.

Colony size data were extracted from the images using the R-package *gitter* (Wagih and Parts, 2014) (used code: *gitter.batch (image.files=filePath, ref.image.file=reference, plate.format = 384, verbose='l', inverse=T)*. Subsequent analyses were performed in R (ver 4.0.3) (R Core Team, 2020). Based on the plate number and row-column position, gene names were assigned to the colony data. Small (<100 pixels) and absent colonies (extreme outliers) were removed in the *ade6Δ* control plates to avoid false positives arising from absent mutants in the mutant library or differences in pinning

and other non-genetic variabilities. These mutants were marked and excluded from subsequent analysis. Then the colony sizes were normalized for spatial effects due to colony position in the plate and plate-to-plate variation by median smoothing and row/column median normalization (Wagih et al, 2013). Any genes within 250 kb of *aal1* or *ade6* genes were excluded from the subsequent analysis as linked loci. For the rest of the genes with mutants in the library, genetic interaction scores (GIS) were calculated as the $\log_{10}$ transformed ratio between the median normalized colony sizes of the experimental query and that of the control. The upper and lower limits of the GIS were arbitrarily set to $+2$ and $-2$ for practical reasons and a cutoff of 0.1 was used to call hits.

## Single-molecule RNA fluorescence in situ hybridization

smRNA-FISH was performed with antisense probes as described (Sun et al, 2020). Since the *aal1* RNA is not detectable during exponential growth and performing smRNA-FISH is technically challenging in ageing stationary phase cells (Ellis et al, 2021), we used the strain overexpressing *aal1* in its native locus (*aal1-gOE*). Cells were grown in EMMG to mid-exponential phase and were fixed in 4% formaldehyde. The cell wall was partially digested using zymolyase. Cells were permeabilized in 70% ethanol, pre-blocked with bovine serum albumin and salmon sperm DNA, and incubated overnight with custom Stellaris oligonucleotide probes (Biosearch Technologies) labelled with CAL Fluor Red 610. Cells were mounted in ProLong Gold antifade mount with DAPI (Molecular Probes), and imaged on a Leica TCS Sp8 confocal microscope, using a 63x/1.40 oil objective. Optical z sections were acquired (0.3 microns z-step size) for each scan to cover the entire depth of cells. The technical error in FISH-quant detection was estimated at 6–7% by quantifying the *rpb1* mRNA foci with two sets of probes labelled with Quasar 670.

## RNA sequencing

Cells were grown in triplicate per genotype and aged in EMMG and sampled at day 6 of CLS. RNA was extracted with the Qiagen RNeasy Mini Kit. Cells were lysed with 0.5 mm acid-washed beads (Sigma) in a FastPrep instrument (MP, FastPrep24-Settings: speed, 6.0 m/s; adaptor, Quick Prep; time, 20 s; ~12 cycles with ≥5 min incubations on ice in between). The amount of beads and the volume of buffer needs to be increased by ~15% and the number of lysis cycles up to ~12 for stationary phase, ageing cells to achieve lysis of >80% of cells. The cells and beads were loosened by flicking the tubes after every 3–4 cycles to increase the efficiency of lysis. RNA was extracted with RNeasy spin columns and digested with DNase I (Qiagen) for 30 min at room temperature followed by a column cleanup with RNeasy spin columns. The rRNA was removed using the Ribo-Zero rRNA removal kit (Illumina) as per recommendations. RNA quality was assessed in an Agilent 2100 Bioanalyzer and strand-specific cDNA libraries were made with NEXTflex™ Rapid Directional qRNA-Seq™ Kit (PerkinElmer, formerly BiooScientific) with molecular indexing. Libraries were quantified in a Qubit, pooled and 4 nM of the pooled library was sequenced in an Illumina HiSeq2500 sequencer, 50 bp paired-end reads in two lanes. All quality control steps including quality filtering, demultiplexing, and adaptor removal were performed with Illumina BaseSpace in-house tools leaving unique molecular indices (UMIs) intact. The quality of the sequences was confirmed with FastQC (ver

0.11.2, https://www.bioinformatics.babraham.ac.uk/projects/fastqc/). Reads for each sample from the two lanes were concatenated and the UMIs were clipped and added to the read header by je (Girardot et al, 2016) (ver. 1.2, http://gbcs.embl.de/je) with the following code: *je clip F1=pair1.fastq.gz F2=pair2.fastq.gz LEN = 8*. Reads were aligned with STAR (Dobin et al, 2013) (ver. 2.6.1a, https://github.com/alexdobin/STAR) against the *S. pombe* genome sequence with the following code: *STAR --genomeDir dir --runThreadN 8 --readFilesIn pair1.fastq.gz pair2.fastq.gz --readFilesCommand gunzip -c --outFileNamePrefix prefix --outSAMtype BAM SortedByCoordinate --quantMode GeneCounts --outWigType wiggle --clip5pNbases 9 --limitBAMsortRAM 2552288998*. Sorted bam files were used to count the number of reads per gene using htseq (Anders et al, 2015) (version 0.12.3, https://htseq.readthedocs.io/en/master/index.html; the code used was: *htseq-count -s reverse -f bam -t gene -i ID file.bam Spombe.gff3 > gene.counts*). The genome sequence and gene annotation gff3 files were downloaded from PomBase (Lock et al, 2019). We performed the differential expression analysis with edgeR (Robinson et al, 2010) (version 3.46.0). Very lowly expressed genes were removed by retaining genes with a mean read count of 1 per million in at least 3 samples. TMM normalization was performed on the filtered counts. Common (overall variability across the genome) and tagwise (measure of the degree of inter-library variation of each gene) negative binomial dispersions were estimated by weighted likelihood empirical Bayes using limma (Ritchie et al, 2015) and fitted into a generalised linear model with strains as a categorical variable with evc (empty-vector control) strain as the reference. Benjamini and Hochberg method was used to control FDRs (False Discovery Rates).

## ChIRP-MS

The standard ChIRP-MS protocol (Chu et al, 2015) was optimized for *S. pombe* cells as described below. Antisense tiling oligo probes for selective pull-down of *aal1* RNA (783 nt) were designed with the online probe designer at *singlemoleculefish.com*, using the following parameters: number of probes = 1 probe/~100 bp of RNA length; target GC percentage = ~45%; oligonucleotide length = 20 nt; spacing length = ~60–80 bp; extensively repeated regions omitted. The probes were checked for homology with other transcripts in the *S. pombe* genome using PomBase Ensembl Blast [options: DNA|DNA database | Genomic sequence | BLASTN | short sequences] and the probes with homology >13 bp to any other region (especially cDNA/RNA) were discarded resulting in 5 usable oligo probes out of 8 designed (Appendix Fig. S1). These antisense DNA probes were synthesized with Biotin-TAGs at the 5′ ends (Sigma) and 100 μM probes were pooled in equimolar ratios. The *aal1-pOE*, evc, *aal1Δ* and wild-type cells were grown in 1-litre cultures each in EMMG until Day 6 after entering stationary phase, monitoring the CLS in 1 ml aliquots sampled every other day. On the Day 6, cells were fixed with 37% formaldehyde (3% final, Sigma) for 30 min at room temperature (RT) with gentle shaking (100 rpm), and the reaction was quenched with 2.5 M glycine (0.125 M final, Sigma) for 10 min at RT. Cells were washed once in ice-cold PBS with PMSF (Sigma, 1 mM final, freshly added before use) and the cell pellets were snap-frozen in liquid-N to store at −80 °C.

Cell pellets were thawed on ice, and resuspended in ice-cold lysis buffer (50 mM Hepes pH 7.6, 1 mM EDTA pH 8.0, 150 mM NaCl, 1% Triton X-100, 0.1% Na-Doc) with freshly added EDTA-free protease inhibitors (Roche), 1 mM PMSF and 100 U/ml SuperaseIn (Invitrogen) in 15 ml tubes. Cells were lysed with 0.5 mm acid-washed glass beads in

a FastPrep (MP Biomedicals, FastPrep24-Settings: 15 ml tube adaptors; speed, – 6.0 m/s; time – 20 s) for 12 cycles with 5 min incubations on ice in between. Supernatants were collected by centrifugation. Cell lysates were transferred to Diagenode 15 ml sonication tubes and sonicated (Bioruptor pico, Diagenode) 30 s ON/45 s OFF at 4 °C for 60 cycles (6 × 10), with vortexing the samples after each 10 cycles. Lysates from the same samples were pooled and centrifuged at $16,000 \times g$ for 10 min at 4 °C to collect the supernatants. Then, the lysates were pre-cleared by incubating with Dynabeads (Invitrogen, magnetic Streptavidin) at 37 °C for 30 min with shaking. Before hybridization, beads were removed twice from lysates. For hybridization, 100 pmol of probes in 2 ml hybridization buffer (750 mM NaCl, 1% SDS, 50 mM Tris-Cl pH 7.0, 1 mM EDTA, 15% formamide), supplemented with protease inhibitors, 1 mM PMSF and SuperaseIn and incubated at 37 °C with shaking overnight. Then, 100 µl Dynabeads per 1 ml lysate were added and incubated at 37 °C for 30 min with shaking. The beads were washed with 5 × 1 ml prewarmed (37 °C) wash buffer (20 mM Tris-HCl pH 8.0, 140 mM KCl, 1.8 mM MgCl$_2$, 10% Glycerol, 0.01% NP-40 with freshly added 1 mM PMSF). During each wash, the beads were incubated at 37 °C with shaking for 5 min. For protein elution, beads were collected on a magnetic stand, resuspended in biotin elution buffer (12.5 mM biotin – Invitrogen, 7.5 mM HEPES pH 7.5, 75 mM NaCl, 1.5 mM EDTA, 0.15% SDS, 0.075% sarkosyl, 0.02% Na-Doc) and incubated at RT for 20 min and then at 65 °C for 10 min with rotation. Eluent were transferred to a fresh tube, and beads were eluted again. The two eluents were pooled, and the residual beads were removed again using the magnetic stand.

TCA was added to 25% of the total volume to precipitate proteins at 4 °C overnight. Then, proteins were pelleted at $16,000 \times g$ at 4 °C for 30 min. The supernatant was carefully removed, the pellets were washed once with cold acetone and air-dried for 1 min. Proteins were immediately solubilized in 8 M urea and tryptic (Promega) digestion was performed overnight at 37 °C with shaking followed by desalting. Peptides were reconstituted in a mixture of 97:3 H$_2$O:acetonitrile (containing 0.1% formic acid). The mobile phase consisted of two components: (A) H$_2$O + 0.1% formic acid and (B) Acetonitrile + 0.1% formic acid. Online desalting of the samples was performed using a reversed-phase C18 trapping column (180 µm internal diameter, 20 mm length, 5 µm particle size; Waters). The peptides were then separated using a linear gradient (0.3 µL/min, 35 °C column temperature), where Buffer A was transitioned from 97% to 60% over 60 min, on an Acquity UPLC M-Class Peptide BEH C18 column (130 Å pore size, 75 µm internal diameter, 250 mm length, 1.7 µm particle size, Waters). The nanoLC system was coupled online with a nanoflow sprayer and connected to a QToF hybrid mass spectrometer (Synapt G2-Si; Waters, UK) to achieve accurate mass measurements using data-independent mode of acquisition (HDMSE) (Patel et al, 2009). Each sample was analysed in technical triplicates, ensuring data reproducibility and reliability. LC-MS grade solvents were used consistently throughout the process: LC-MS H$_2$O (Pierce), LC-MS Grade Acetonitrile (Pierce) and LC-MS Formic Acid (Sigma). Lockmass calibration was performed using [Glu1]-fibrinopeptide B (GFP, Waters) at a concentration of 100 fmol/µL. The lockmass solution was introduced through an auxiliary pump at a flow rate of 0.5 µL/min to a reference sprayer, which was sampled every 60 sec.

The acquired data was processed using PLGS v3.0.2 (Waters). The data was queried against an *S. pombe* FASTA protein database (UniProt proteome: UP000002485) concatenated with a list of common contaminants obtained from the Global Proteome Machine (ftp://ftp.thegpm.org/fasta/cRAP). The identification parameters included Carbamidomethyl-C as a fixed modification, and Oxidation (M) and Phosphorylation of STY as variable modifications. To account for incomplete digestion, a maximum of two missed cleavages were allowed. Peptide identification required a minimum of 3 corresponding fragment ions, while protein identification necessitated a minimum of 7 fragment ions. The protein false discovery rate was set at 1%. All identified proteins were tabled in Dataset EV2. Differential protein enrichment analysis was performed with DEP (Zhang et al, 2018). As we observed that imputation of missing values with any available method in the package resulted in false positives, we adopted the following strategy. Any protein not identified in at least 2 out of 3 replicates of at least 1 strain was removed resulting in 218 proteins being identified. The remaining missing values were imputed with 1. The data was background corrected and normalized by variance stabilizing transformation. A stringent cut-off of FDR ≤ 0.005 and $\log_2 \geq 6.5$ in *aal1-pOE* and/or wild type relative to *aal1Δ* was applied to eliminate *aal1*-RNA-independent background interactions.

## ChIRP-seq

ChIRP-seq samples were identical to those described in ChIRP-MS and the washed beads were reconstituted in equal volume of proteinase K RNA buffer (100 mM NaCl, 10 mM TrisCl pH 7.0, 1 mM EDTA, 0.5% SDS) and reverse cross-linked at 70 °C for 1 h with end-to-end shaking followed by another 1 h incubation at 55 °C after freshly adding proteinase K (Ambion, 5% by volume from 20 mg/ml). Then, the samples were boiled for 10 min at 95 °C. RNA was extracted with QIAzol and RNeasy Mini kit (Qiagen) according to the manufacturer's recommendations, including a DNase treatment. Sequencing libraries were prepared with NEXTflex Rapid Directional qRNA-Seq Kit (PerkinElmer), spiked with 20% PhiX (Illumina) to increase the complexity of the libraries to accommodate clustering and sequenced in an Illumina MiSeq (75 bp, paired-end) instrument. The read processing was as described for RNA-Seq. Reads were mapped with Hisat2 (Kim et al, 2019) (version 2.2.1, code used: *hisat2 -x genomeIndex --rna-strandness FR --trim5 8 -1 pair1.fastq.gz -2 pair2.fastq.gz -S out.sam --summary-file summary.txt*). sam files were converted to bam files and sorted/indexed with samtools (ver 1.14) (Danecek et al, 2021) and sorted bam files were used to count reads per gene with htseq (Anders et al, 2015) (ver. 0.12.3, https://htseq.readthedocs.io/en/master/index.html; code use: *htseq-count -s reverse -f bam -t gene -i ID file.bam annotation.gff3 > file.count*). The top *aal1*-bound RNAs were determined with edgeR (Robinson et al, 2010) (version 3.32.1; Dataset EV4) from two replicates each of *aal1Δ*, wild type (wt, 972h-) and *aal1-pOE* strains grown and processed independently. Since there were no significantly enriched RNAs (FDR ≤ 0.05) in the wild-type relative to *aal1Δ* cells, we chose the top 30 enriched RNAs which were verified in IGV (Thorvaldsdottir et al, 2013) (ver 2.8.0, https://software.broadinstitute.org/software/igv/). For IGV inspection, strand-specific MiSeq reads were processed further with deepTools (Ramirez et al, 2016) (ver 3.5.1, https://deeptools.readthedocs.io/en/develop/) to get normalized gene coverage in counts per million (cpm) per 50 bp bins (bam to bigWig files) for direct comparison.

## In silico prediction of *aal1-rpl1901* lncRNA-mRNA interaction

In silico predictions of *aal1RNA-rpl1901mRNA* interactions were performed with the ViennaRNA Package (Lorenz et al, 2011) (ver. 2.5.0, https://www.tbi.univie.ac.at/RNA/documentation.html) with RNAcofold (Lorenz et al, 2016). RNAcofold tests the probability of RNA-RNA interactions of the provided RNA pairs by computing minimum free energy structures and a base pairing probability matrix. Potential RNA secondary structures were considered when the pairwise base pairing probabilities and probable interaction sites were predicted. Similarly, the concentration dependency of homo- and hetero-interactions of RNAs were computed for the given input concentrations of the monomers (in mol/lit) and presented as concentration dependency plots of RNA-RNA interactions.

## Puromycin incorporation assay

*S. pombe* cells were grown with constant shaking (180 rpm) at 32 °C to mid-exponential phase in 25 ml EMM2 media. Puromycin (Gibco, #A11138-02) was added to a final concentration of 10 µM and incubated at 32 °C for 30 min with shaking. Cells were harvested and snap-frozen in liquid nitrogen. Pellets were thawed on ice and broken with beads (Sigma, G8772) in protein lysis buffer (50 mM Tris, pH 7.5, 150 mM NaCl, 5 mM EDTA, 10% glycerol, 1 mM PMSF and protease inhibitor cocktail [cOmplete™ Mini EDTA-free, Roche]). Approximately 10 µg lysate was mixed with Laemmli buffer with freshly added 100 mM DTT and boiled at 95 °C for 5 min. Proteins were separated on a gradient gel (Thermo Fisher Scientific, #NP0322BOX) and transferred onto a nitrocellulose membrane (Amersham) for 30 min using a semidry transfer system (Trans-Blot Turbo, BioRad). We used anti-puromycin antibody (Millipore, #12D10, 1:2500), HRP conjugated anti-mouse secondary (Abcam, #ab6789, 1:10,000) and anti-beta-actin antibody (ab8224, Abcam, 1:10,000). Blots were developed with Luminata Forte Western substrate (Millipore) and imaged in an Amersham ImageQuant800 imager. The intensity of anti-puromycin bands between 15 and 165 kDa was quantified from chemiluminescence images with the ImageQuant™ TL 10.2 analysis software with rolling ball background normalization. Expression was calculated relative to actin and compared to the respective controls. Quantification data was analysed with a linear regression model which includes genotype and batch as variables and the significance determined with t-test.

## Cloning and generation of *UAS-aal1* fly line

The *aal1* sequence was PCR amplified from *S. pombe* including the predicted 3′ polyadenylation signal sequence. Using overlap-extension PCR, *UAS*, and *Drosophila HSP-70* promoter sequences were added at the 5′ end, and an additional 150 bp of SV40 polyadenylation signal sequence was added at the 3′ end. The resulting product was transferred to the *pCaSpeR* plasmid by double digestion with *BamHI* and *Not1*, followed by ligation with T4 DNA ligase, and transferred into competent *Escherichia coli*. The correct clone was confirmed by DNA sequencing at Source Biosciences. The plasmid miniprep of the clone was injected into *white^1118* embryos and randomly integrated into the fly genome using piggyBac transgenesis (Gregory et al, 2016), at the Department of Genetics Fly Facility (University of Cambridge, https://www.flyfacility.gen.cam.ac.uk/Services/Microinjectionservice/). The

*UAS-aal1* fly line with balancer chromosomes was then provided by the Cambridge fly facility. To confirm whether *aal1* was expressed in the fruit fly, we crossed the *UAS-aal1* homozygous males with *daughterless GAL4* virgins. The flies emerging from this cross were collected and RNA was extracted from five 7-day-old adult flies, using TRIzol (Invitrogen) according to the manufacturer's protocol. cDNA was synthesized using random hexamers and Superscript II (Invitrogen) according to the manufacturer's instructions. Using this cDNA as a template, the expression of *aal1* RNA was then confirmed by PCR using primers listed in Appendix Table S1.

## Fly husbandry and maintenance

We used an outbred wild-type stock that was initially collected from Dahomey (present Benin) in 1970 and subsequently maintained in large population cages on a 12 h:12 h light/dark cycle at 25 °C to maintain lifespan and fecundity at levels similar to wild-caught flies. The *white^1118* mutation was introduced into this background to allow easier tracking of transgenes and Wolbachia infection was cleared by tetracycline treatment. Before the experiments, the *aal1* fly lines and gene-switch drivers (TIGS) were backcrossed into this white Dahomey (wDah) background for at least six generations. All stocks were maintained, and experiments were conducted at 25 °C and 60% humidity with 12 h:12 h light/dark cycles, on SYA food containing 10% brewer's yeast, 5% sugar, and 1.5% agar with nipagin and propionic acid added as preservatives.

## Lethality test in flies

To test whether *aal1* exhibits lethality when expressed during development, we crossed *UAS-aal1* homozygous males with daughterless *GAL4* virgins. The adults were allowed to mate for 48 h, and eggs were collected within 24 h and counted. After 10 days, the number of flies that emerged from the resultant crosses was recorded. We used GAL4-alone and UAS-alone controls in parallel.

## Lifespan analysis in flies

For lifespan assays, experimental flies were generated from suitable crosses in cages containing grape juice, agar, and live yeast. Flies were allowed to mate and the eggs were collected after 22 h and 20 µL of egg sediments (in 1xPBS) were seeded on SYA medium in glass bottles to rear flies at standardized larval densities. Flies emerged after 10 days, and were transferred to new bottles where they were allowed to mate for 48 h before sorting females into experimental vials at a density of 15 flies per vial. To induce transgene expression using the GAL4/UAS GeneSwitch system, RU486 (Sigma, dissolved in ethanol) was added to the media at a final concentration of 200 µM. As a control (RU-), equivalent volumes of the vehicle alone were added. Flies were transferred to fresh vials three times a week and their survival scored. To control for potential $RU_{486}$ artefacts, driver-only controls feeding $RU_{486}$ were included in the experiment.

# Data availability

ChIRP-MS data have been submitted to the ProteomeXchange Consortium via the PRIDE partner repository with the dataset identifier PXD045625: http://www.ebi.ac.uk/pride/archive/projects/

PXD045625. ChIRP-seq and RNA-seq data have been submitted to GEO under the accession number GSE243036: https://www.ncbi.nlm.nih.gov/geo/query/acc.cgi?acc=GSE243036.

The source data of this paper are collected in the following database record: biostudies:S-SCDT-10_1038-S44319-024-00265-9.

## Peer review information

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

## Acknowledgements

We thank Melania D'Angiolo and Manuel Lera-Ramirez for their helpful comments on the manuscript. This work was supported by the Biotechnology and Biological Sciences Research Council (Grant numbers BB/R018219/1 to JB and BB/W013525/1 to NA), the Medical Research Council (to SM), and a Bangabandhu Overseas Scholarship, University of Dhaka, to AFS.

## Author contributions

**Shajahan Anver**: Conceptualization; Resources; Data curation; Formal analysis; Validation; Investigation; Visualization; Methodology; Writing—original draft; Writing—review and editing. **Ahmed Faisal Sumit**: Investigation; Methodology. **Xi-Ming Sun**: Methodology. **Abubakar Hatimy**: Methodology. **Konstantinos Thalassinos**: Supervision; Funding acquisition; Methodology; Writing—review and editing. **Samuel Marguerat**: Supervision; Funding acquisition; Investigation; Methodology; Writing—review and editing. **Nazif Alic**: Supervision; Funding acquisition; Investigation; Methodology; Writing—review and editing. **Jürg Bähler**: Conceptualization; Formal analysis; Supervision; Funding acquisition; Investigation; Visualization; Writing—original draft; Project administration; Writing—review and editing.

Source data underlying figure panels in this paper may have individual authorship assigned. Where available, figure panel/source data authorship is listed in the following database record: biostudies:S-SCDT-10_1038-S44319-024-00265-9.

## Disclosure and competing interests statement

The authors declare no competing interests.

# Expanded View Figures

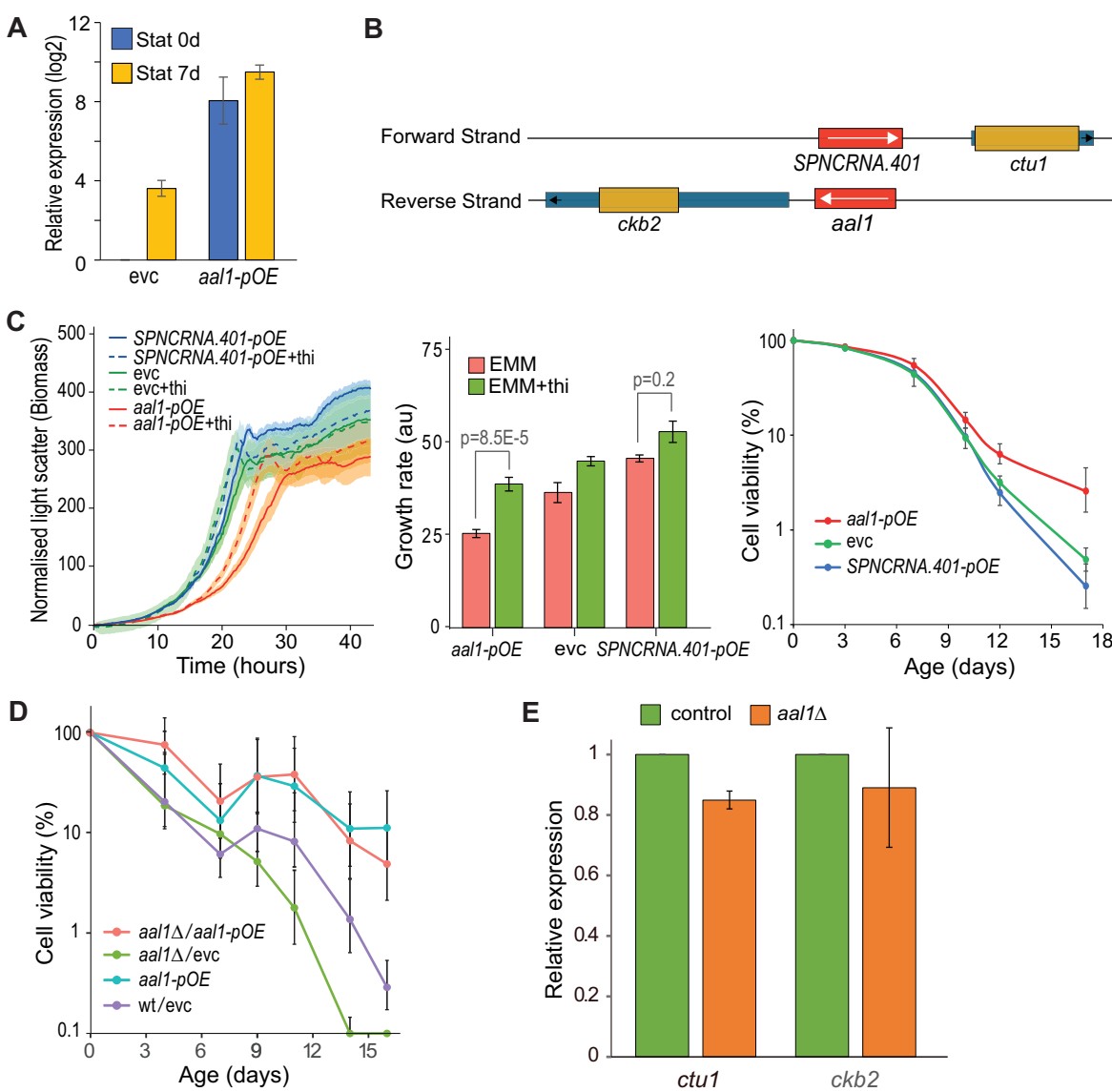

**Figure EV1. Analysis of the *SPNCRNA.401* lncRNA.**

(A) Expression of *aal1* in *aal1-pOE* cells at the onset of stationary phase (Stat 0d) and after 7 days in stationary phase (Stat 7d) relative to empty-vector control cells (evc) at the onset of stationary phase (Stat 0d), measured with strand-specific RT-qPCR. The *aal1* RNA levels are normalized to the lowly expressed coding gene *ppb1*. Bars indicate the mean±SD (standard deviation) of three independent repeats. (B) Genomic environment of *aal1* gene showing a 745 nucleotide overlap in antisense direction with the *SPNCRNA.401* gene. *Red boxes*: lncRNA genes, with the transcriptional direction indicated by white arrows; *ochre and blue boxes*: open reading frames and untranslated regions, respectively, of coding genes. Visualization using the PomBase genome browser (Harris et al, 2022). (C) Left graph: Growth assay for cells ectopically overexpressing *aal1* and *SPNCRNA.401* under the thiamine-repressible *P41nmt1* promoter (*aal1-pOE* and *SPNCRNA.401-pOE*) compared to empty-vector control (evc) with/or without 15 µM thiamine (thi) added to the medium. Experimental setup and analysis as in Fig. 1D. Middle graph: Quantitation of growth rate for experiments shown in the left graph, as in Fig. 1D. Right graph: CLS assays for *aal1-pOE* and *SPNCRNA.401-pOE* cells compared to empty-vector control (evc) cells. Experimental setup and analysis as in Fig. 1C. (D) CLS assays for *aal1Δ* cells ectopically overexpressing *aal1* (*aal1Δ/aal1-pOE*) compared to *aal1Δ* and wild-type cells overexpressing empty-vector controls (*aal1Δ/evc; wt/evc*) and *aal1-pOE* cells. (E) Expression of genes flanking *aal1* in the presence and absence of *aal1*. RT-qPCR experiment to determine transcript levels of *ctu1* and *ckb2* in *aal1Δ* cells relative to wild-type cells (control). Four independent biological repeats were carried out using 7-day-old stationary-phase cells. Data were normalized to *act1* expression. Expression of *ctu1* was slightly lower in *aal1Δ* compared to control cells ($p_{Student's\ T}$ ~ 0.001), while expression of *ckb2* showed no significant difference ($p_{Student's\ T}$ ~ 0.35).

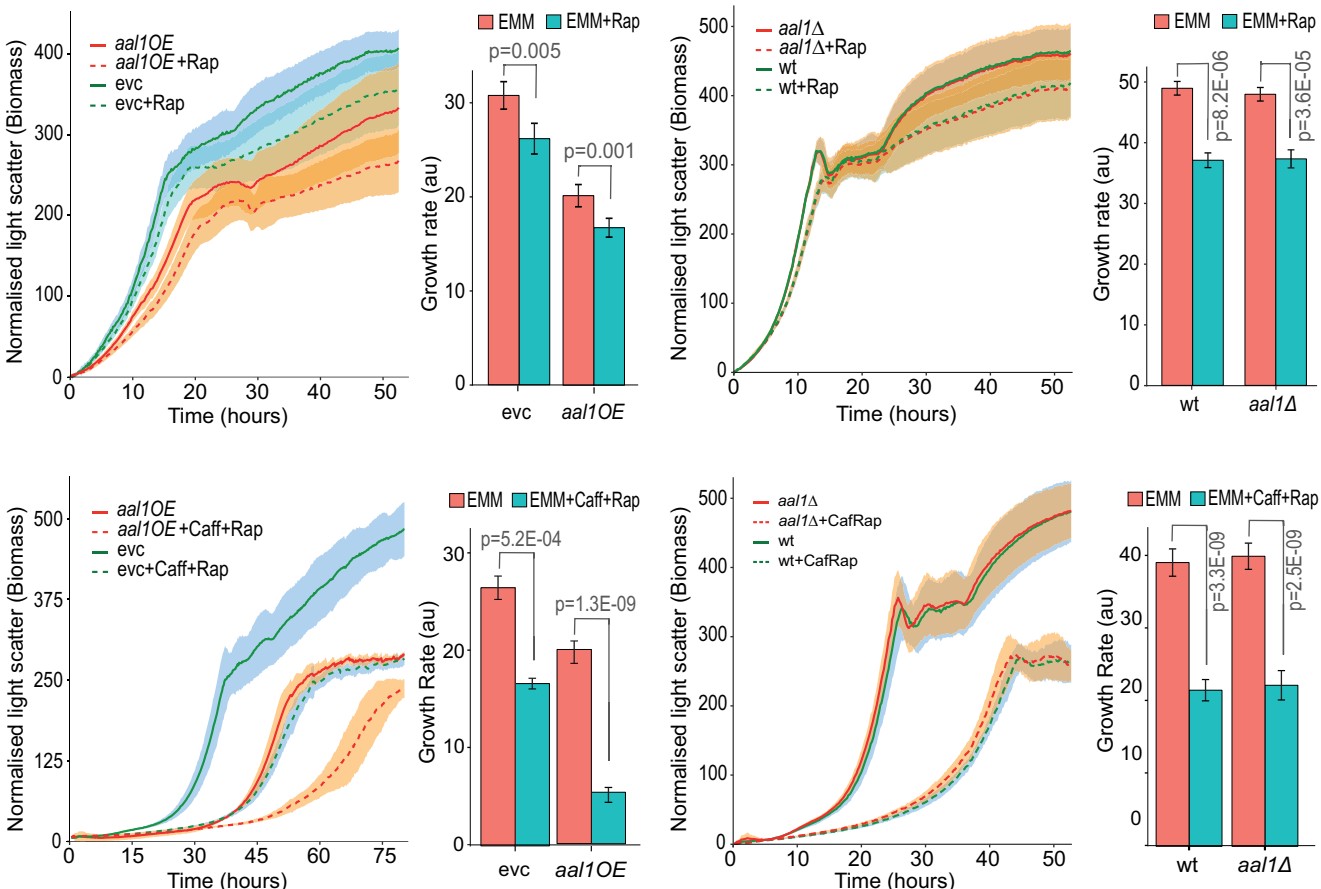

**Figure EV2.  *aal1* phenotypes do not depend on TORC1 signalling.**

Top graphs: Rapamycin inhibits cell growth in *aal1-pOE* and *aal1Δ* mutants to a similar degree as in the respective controls, indicating that TORC1 and *aal1* functions exert additive effects. Rapamycin (300 ng/ml) was added after 8 h of initial growth to avoid overly long lag periods. Bottom graphs: The combination of caffeine (10 mM) and rapamycin (100 ng/ml) leads to a stronger inhibition of cell growth in *aal1-pOE* and *aal1Δ* mutants, similar as in the respective controls, indicating again that TORC1 and *aal1* functions have additive effects. Cells were grown in a microbioreactor and mean growth curves were fitted with *grofit* (Kahm et al, 2010), with SD shown as shades. Quantitation of growth rates (mean ± SE) for experiments is shown in the bar graphs. Details as in Fig. 1D,E.

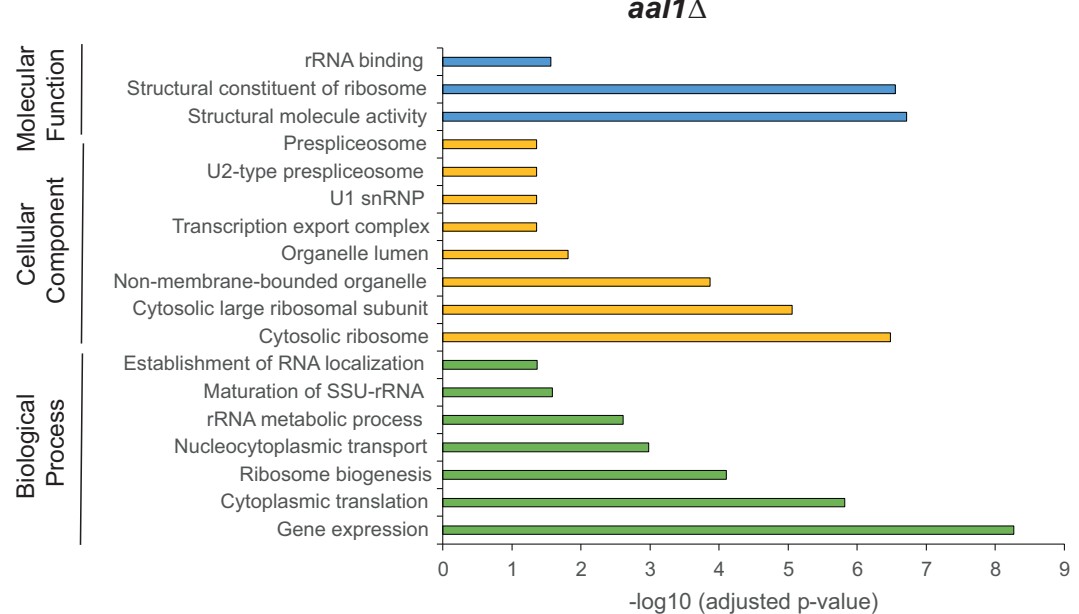

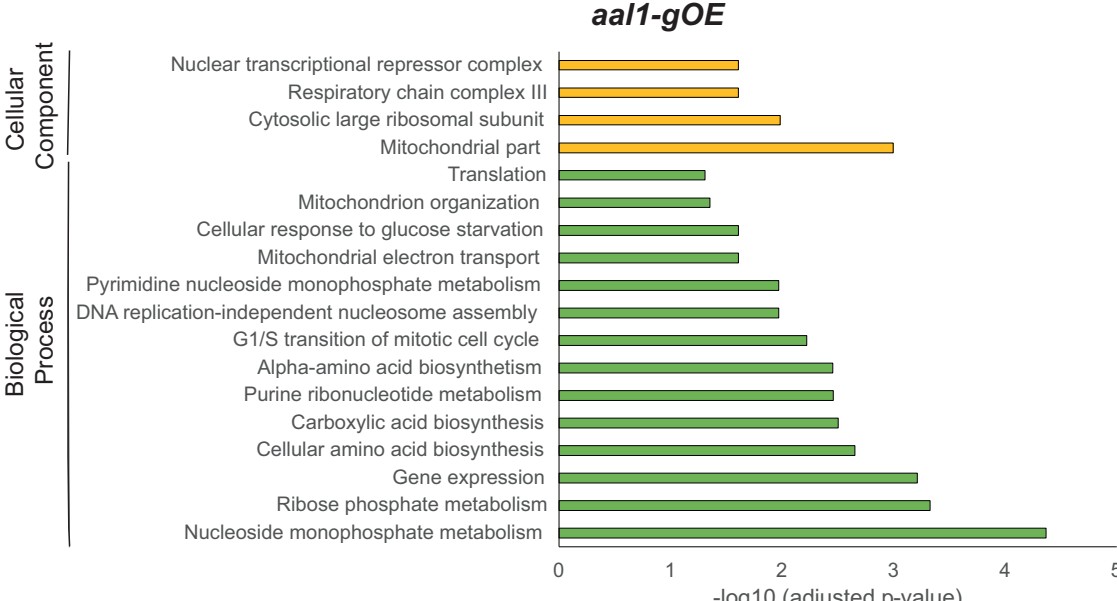

**Figure EV3. Functional enrichments among genes genetically interacting with *aal1Δ* and *aal1-gOE* mutants.**

GO-term enriched among the genes that showed positive or negative genetic interactions (adjusted *p*-value[FDR] ≤0.05) in at least 2 of the 3 repeats in the SGA screens using *aal1Δ* (top) or *aal1-gOE* (bottom) as query mutants (see Methods). Representative GO terms for Biological Process, Molecular Function, and Cellular Component are shown, selected for non-redundancy, specificity, and significance. The graphs show the $-\log_{10}$ of adjusted *p*-values (false-discovery rate) for enrichment of the different terms. Visualisation with ShinyGO (ver 0.77). The genetic-interaction and background gene lists are provided in Dataset EV1.

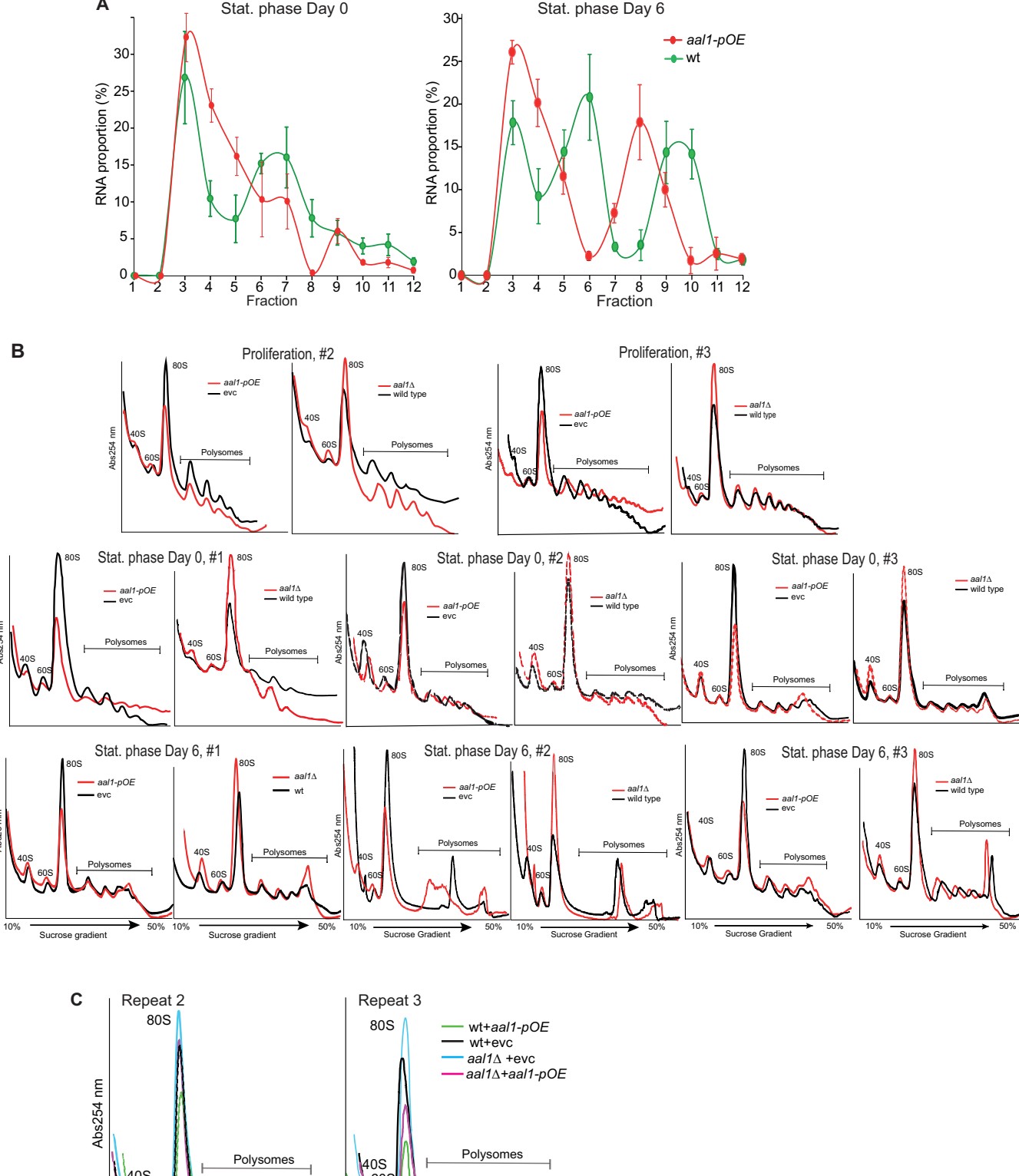

◄

**Figure EV4.** *aal1* **associates with ribosomes and reduces the cellular ribosome content.**

(A) Polysome fractionation followed by RT-qPCR shows that *aal1* binds to ribosomes during early stationary phase (Day 0, left graph) and late stationary phase (Day 6, right graph) in both wild-type (green) and *aal1-pOE* (red) cells. (B) Independent biological repeats of polysome profiling as in Fig. 4A,B for proliferating cells (top), early stationary-phase cells (middle) and late stationary-phase cells (bottom) for the four strains indicated. The two profiles are aligned at the lowest points of the monosome peaks, corresponding to the baseline. (C) Two independent biological repeats of experiment shown in Fig. 4C.

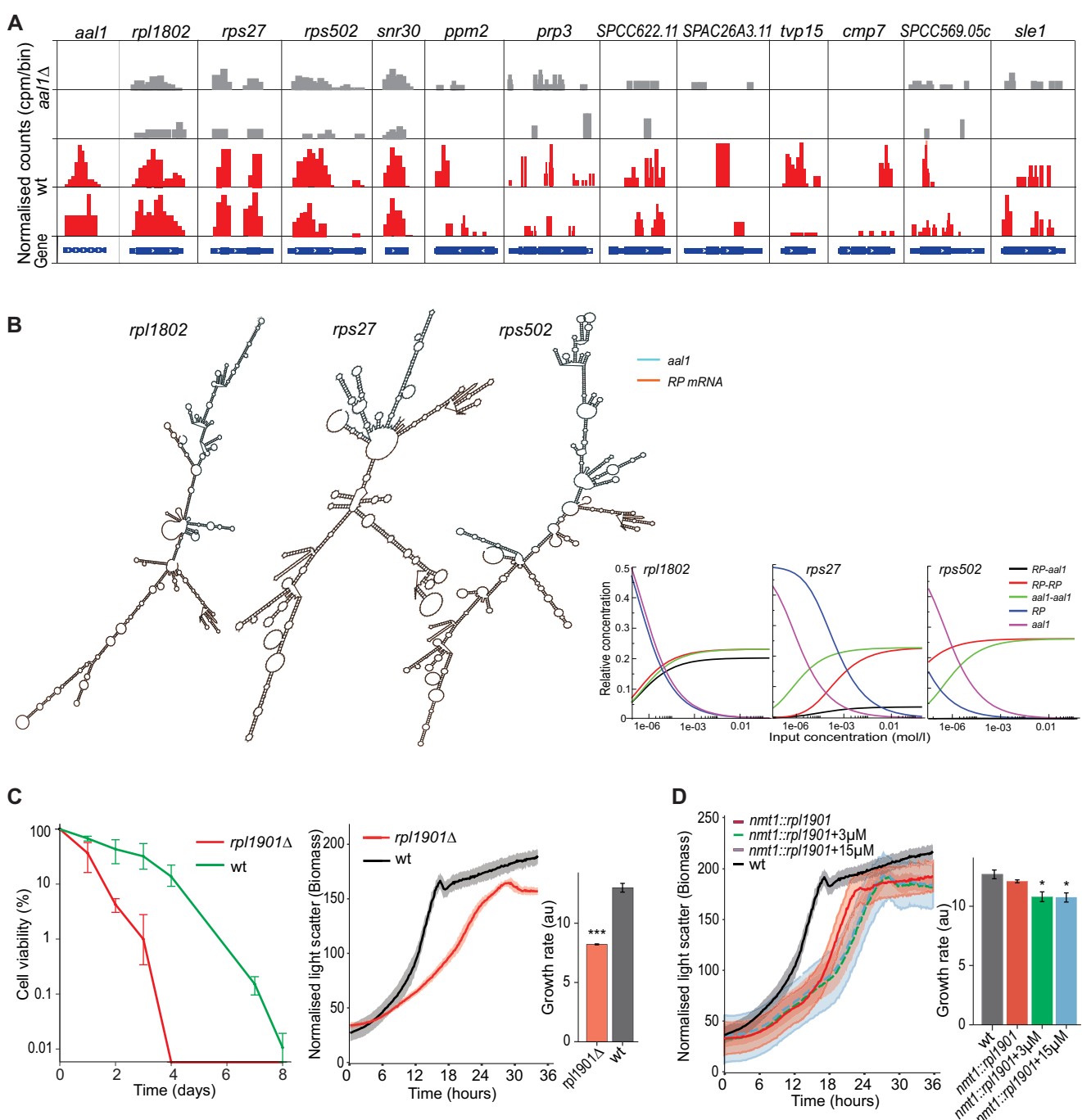

Figure EV5. Analyses of RNAs binding to *aal1*.

(A) IGV tracks from strand-specific ChIRP-seq reads in counts per million (cpm) per 50 bp bins (deepTools) (Ramirez et al, 2016) across *aal1* (control) and prospective target RNAs as indicated on top. Top *aal1*-bound RNAs were determined with edgeR (Robinson et al, 2010) (Dataset EV4) from two replicates each of *aal1Δ*, wild type (wt) and *aal1-pOE* cells, and the data were verified in IGV (Thorvaldsdottir et al, 2013) (details in Methods). The prospective *aal1*-bound RNAs in the ChIRP-seq data include three additional mRNAs encoding ribosomal proteins and one small nucleolar RNA (*snr30*). (B) In silico predictions of interaction between *aal1* and *rpl1802, rps27,* and *rps502* using the ViennaRNA package (Lorenz et al, 2011) with RNAcofold (Lorenz et al, 2016). Left: Predicted *aal1-RP* (ribosomal protein) mRNA heterodimers with interaction sites and potential RNA secondary structures. Right: Concentration dependency plots of dimerization showing the computed homo- and hetero-dimerizations of RNAs for concentration relative to each other (y-axis) and different input concentrations (x-axis), with predicted equilibrium concentrations for the monomers, homodimers, and heterodimers as indicated. (C) Left graph: Chronological lifespan assays for *rpl1901Δ* and wild-type cells, performed in rich medium. Right graphs: Growth assays of *rpl1901Δ* and wild-type cells and quantitation of growth rate for these assays. Experimental setup and analysis as in Fig. 1D. Statistical significance was determined with one-way ANOVA followed by Dunnett's test (Hothorn et al, 2008), with $p < 0.0001$ relative to wt. (D) Growth assays of *nmt1:rpl1901* and wild-type cells with the addition of different doses of thiamine as indicated and quantitation of growth rate for these assays. Experimental setup and analysis as in Fig. 1D. Statistical significance was determined with one-way ANOVA followed by Dunnett's test (Hothorn et al, 2008), with $p < 0.006$ relative to wt.

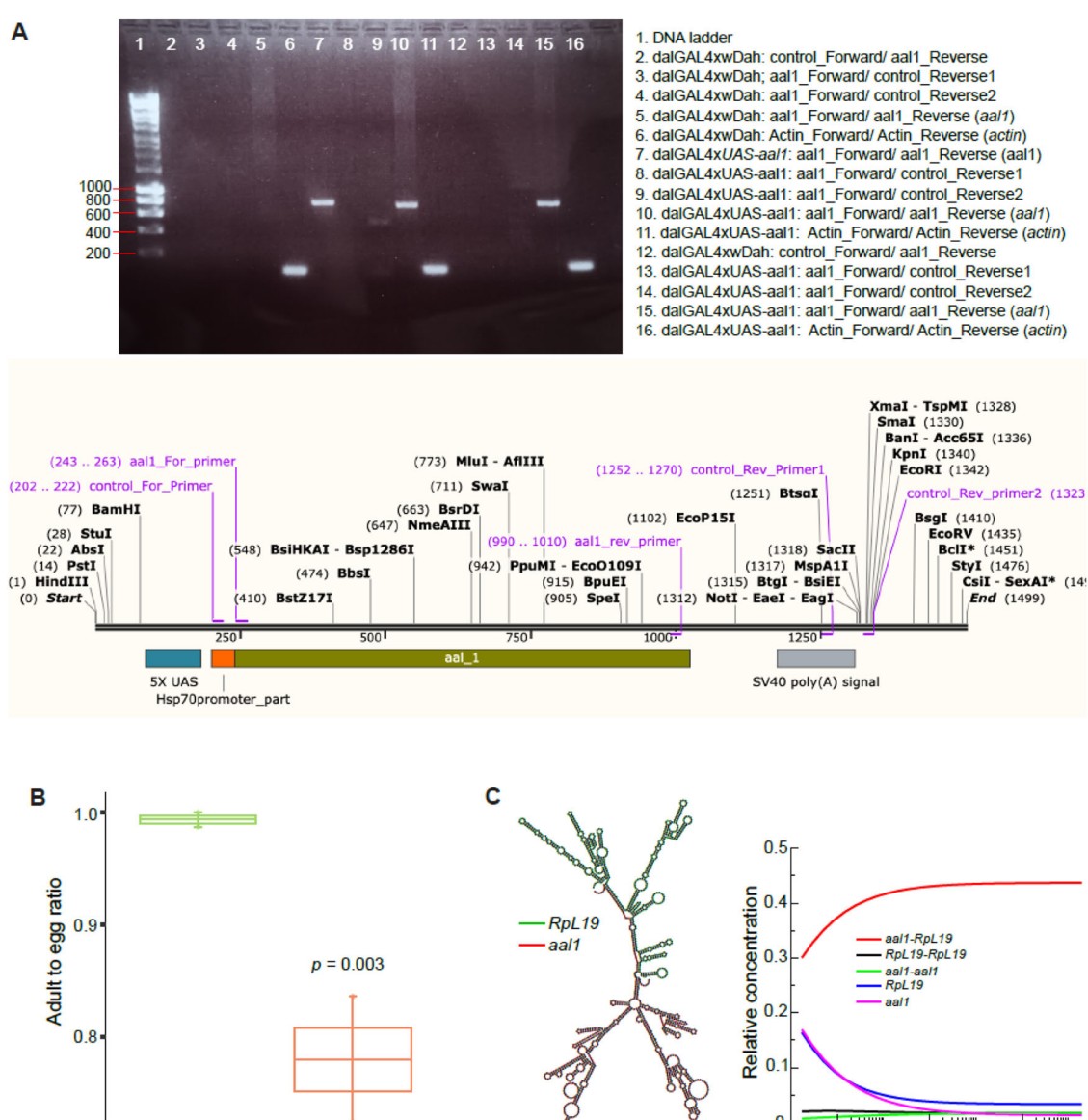

**Figure EV6. Supporting analyses for experiment expressing *aal1* in flies.**

(A) Top: Confirmation of *aal1* expression in the inducible *UAS-aal1* strain flies. The expression of an RNA of expected size (783 nt) was confirmed in females where *UAS-aal1* was driven by the ubiquitous, constitutive GAL4 driver (*daughterlessGAL4/dalGAL4*). Random primed cDNA was used as template. Primer positions as follows (see scheme). Forward control: resides in Hsp70 promoter; Reverse_control1: in SV40 polyA signal; Reverse_control2: in downstream sequence of SV40 polyA signal; aal1_Forward & aal1_Reverse: in 5′ and 3′ ends of *aal1* transcript, respectively; Actin_Forward/Actin_Reverse: housekeeping Actin gene. wDah control crossed to dalGAL4 was used as a negative control strain (Lanes 2–6 and 11). Three *UAS-aal1* replicates were tested (lanes 7, 8–11, 13–16). Primer sequences are provided in Appendix Table S1. Bottom: Scheme of *UAS-aal1* construct showing the positions of the primers used (purple), visualized with SnapGene Viewer 5.3.2. (B) Ubiquitous expression of *aal1* in flies with a *dalGAL4* promoter throughout development significantly reduces the number of flies that reach adulthood (details in Methods: Lethality test in flies). Statistical significance determined with two-sample t-test. (C) In silico prediction of interaction between *S. pombe aal1* and *Drosophila RpL19* using the ViennaRNA package (Lorenz et al, 2011) with RNAcofold (Lorenz et al, 2016). Left: Predicted *aal1-RpL19* heterodimer structure showing the interaction sites along with potential RNA secondary structures. Right: Concentration dependency plot of dimerization showing the computed homo- and hetero-dimerizations of RNAs for concentration relative to each other (y-axis) and different input concentrations (x-axis), with predicted equilibrium concentrations for the two monomers, *aal1* and *RpL19*, the two homodimers, *aal1-aal1* and *RpL19-RpL19*, and the *aal1-RpL19* heterodimer.

