## [Peer Review File · EMBO Reports]

Ageing-associated long non-coding RNA extends lifespan and reduces translation in non-dividing cells

Shajahan Anver, Ahmed Sumit, Xi-Ming Sun, Abubakar Hatimy, Konstantinos Thalassinos, Samuel Marguerat, Nazif Alic, and Jürg Bähler

Corresponding author(s): Jürg Bähler (j.bahler@ucl.ac.uk)

Review Timeline:	Transfer Date:	21st Jul 24
	Editorial Decision:	21st Aug 24
	Revision Received:	7th Sep 24
	Accepted:	11th Sep 24

Editor: Esther Schnapp

Transaction Report: This manuscript was transferred to

EMBO reports following peer review at Review Commons.

**Review
COMMONS**

Review #1

1. Evidence, reproducibility and clarity:

Evidence, reproducibility and clarity (Required)

This manuscript addresses the function of a lncRNA in yeast that the authors name *aal1* (aging-associated lncRNA) that is induced in quiescent cells. They find that deletion of *aal1* shortens chronological lifespan, while overexpression prolongs lifespan and these effects are correlated with changes in the levels of ribosomal proteins. Unbiased mapping of epistatic interactions confirms this important functional relationship between *aal1* and the ribosome, as does ChIRP-MS approaches that identified RBPs associated with the *aal1* RNA. The authors in particular find that *rpl1901* mRNA, which encodes a ribosomal protein, is a potential direct binding target of *aal1* and the levels of this mRNA are accordingly modulated upon changing the levels of *aal1*. It is thus proposed that *aal1* acts by decreasing *Rpl1901* levels, leading to reduced ribosome content that attenuates protein translation and promotes longevity. Amazingly, overexpressing the *aal1* lncRNA is sufficient to extend lifespan in flies, despite the fact that *aal1* is not evolutionarily conserved in flies. This suggests a conserved underlying mechanism across species that this lncRNA is acting upon. Overall, the authors have done a nice job of using a variety of high-throughput techniques to define a clear molecular mechanism, which they have validated with supporting experiments.

1. Fig 1C: Please clarify the degree that *aal1* is over-expressed in the over-expressed cells.
2. Fig 2B: To confirm that the overexpressed *aal1* transcript accurately recapitulates endogenous localization, the authors should consider performing nuclear cytoplasm biochemical fractionation followed by QPCR to address if endogenous *aal1* does shown enrichment in the cytoplasm.
3. Fig 4C: This figure should be zoomed in and more details provided so that the degree of potential base pairing is clear. It is impossible to tell from the data presented and there is minimal description in the text.
4. To really prove the base pairing model in Fig 4C, it would be really fascinating to overexpress a mutated form of *aal1* that can not base pair as efficiently to *Rpl1901* mRNA and see if the phenotypes are lost.

****Minor points:****

1. Page 3: For completion, it would be useful to mention the known functions, if any, of the *ckb2* and *ctu1* genes that are neighboring *aal1*.

2. Significance:

Significance (Required)

This manuscript addresses a novel role for a lncRNA in controlling longevity in yeast and shows that even though the lncRNA itself is not conserved in flies, forced overexpression of it in flies is sufficient to extend lifespan. I think this opens up new avenues for understanding aging related pathways and will be of interest to researchers in both the RNA and aging communities.

3. How much time do you estimate the authors will need to complete the suggested revisions:

Estimated time to Complete Revisions (Required)

(Decision Recommendation)

Between 1 and 3 months

4. Review Commons values the work of reviewers and encourages them to get credit for their work. Select 'Yes' below to register your reviewing activity at Web of Science Reviewer Recognition Service (formerly Publons); note that the content of your review will not be visible on Web of Science.

Yes

Review #2

1. Evidence, reproducibility and clarity:

Evidence, reproducibility and clarity (Required)

In their manuscript "Aging-associated long non-coding RNA boosts longevity and reduces the ribosome content of non-dividing fission yeast cells", Anver et al. explore the biological function of a previously uncharacterised lncRNA, SPNCRNA.1530 (termed aging-associated lncRNA, or aal1, based on the findings in this manuscript), which had been identified in a previous study from the same lab as one of four lncRNAs that significantly slowed down growth when overexpressed. Aal1 levels are restricted by nuclear decay enzymes in proliferating cells, but strongly increase (> 15-fold) in stationary cells. Using a deletion strain and overexpression constructs, the authors demonstrate that aal1 levels correlate with increased chronological live span and slower growth. This is also true when aal1 is expressed from an ectopic location, suggesting that the lncRNA functions in trans. Aal1 localizes to the cytoplasm when overexpressed, and SGA analysis of the deletion strain revealed widespread genetic interaction with genes involved in protein translation. Pulldown of aal1 with antisense oligos enriched ribosomal proteins and several highly expressed metabolic enzymes, in addition to various ncRNAs and mRNAs. Four of the interacting mRNA encoded for ribosomal proteins, among them rpl1901 mRNA, for which dimer formation with aal1 could be predicted. The authors suggest a model where aal1 modulates levels of ribosomal proteins, potentially through direct interaction with rpl1901 mRNAs, leading to decreased cellular ribosome content and increased chronological life span. In support of this model, they demonstrate that varying rpl1901 levels with the help of a tunable promoter also correlated with changes in life span changes. Finally, they also showed that expression of fission yeast aal1 in the *Drosophila* gut was able to extend life span of the fly.

****I raise the following major points:****

1. I do not find the data supporting the interaction of aal1 with ribosomes particularly convincing.

Yes, there is sedimentation of the ncRNA on polysome gradients, but the overlap with different ribosomal fractions is not very clear. Also, the pattern of ncRNA sedimentation is very different between wt and OE strain (S4A), whereas the polysome profile shows only little change. Other factors, including condensation of aal1 in ribonucleoprotein condensates (as the authors suggest) could also influence sedimentation behaviour. Disrupting ribosomes in the gradients and observation of a co-shift of aal1 could help to support a direct interaction.

The ChIRP-MS experiment identified many ribosomal proteins but also many other highly expressed proteins, including act1, adh1 and pma1. Setting this in relation to protein expression data could be helpful. However, as I understand it, the model of rpl1901 repression by direct interaction with aal1 that the authors suggest does not require an interaction with ribosomal proteins or ribosomes to be functional.

2. There must be a better way of quantifying cellular ribosome content than comparing total absorption in polysome profiles. In particular, as the authors state in the methods section "The lysates were quantified in a Nanodrop (OD260). Equal amounts of each lysate were loaded for the polysome fractionation." A260 measurements are dominated by ribosomal RNA. How can this be quantitative? One assay that has been used to determine translation capacity is puromycin labelling (but may not be straightforward in yeast).

****Minor points:****

1. Resolution of some of the figures is poor in the submitted pdf. In Fig 4C, this makes it impossible to distinguish both RNAs in the predicted dimer structure. Please check. Also, please use a colour combination different than red and green to make the images accessible to people who are colour-blind.

2. Have the authors run structure predictions with the three other ribosomal protein mRNAs that were detected in ChIRP? Some of these were also very significantly enriched. Could the potential for dimer formation with aal1 be conserved among these?

3. At times it is not very clear to me why a particular strain was chosen for a particular experiment. For the SGA analysis, the authors use the deletion strain to test impact on growth under proliferating conditions. Considering that expression of aal1 is almost fully repressed in the wt under these conditions, the impact of the deletion predictably is very low. Using the overexpressing strain (or an assay that tests stationary conditions) might have resulted in stronger or more genetic interactions.

Similarly, the differential gene expression experiment is carried out in stationary cells, but here the authors compare the wt (where aal1 is derepressed under these conditions) to the overexpressing strain. Here, comparing to the deletion strain may have rounded off the experiment.

****Referees cross-commenting****

Re Reviewer #1's comment (2)

"(2) Fig 2B: To confirm that the overexpressed aal1 transcript accurately recapitulates endogenous localization, the authors should consider performing nuclear cytoplasm biochemical fractionation

followed by QPCR to address if endogenous aal1 does shown enrichment in the cytoplasm."
My comment: Biochemical nuclear-cytoplasmic fractionation is notoriously difficult in yeast, so this may be a hard experiment to do.

Re: Reviewer #3` s comment (1):

"1. It is not clear whether overexpression of aal1 affects transcription (inferred from RNA seq) or translation of Rs genes in aging cells? "

My comment: In my understanding, the model is that it affects RNA levels, but through a post-transcriptional mechanism.

2. Significance:

Significance (Required)

The link between aal1 expression and chronological life span is well established and strongly supported by the data provided. An involvement in the regulation of protein translation is supported by the genetic interaction data as well as the differential gene expression analysis in the overexpressing strain, and highly plausible in the light of other findings in the ageing field. A possible mechanism is also put forward, suggesting that dimerization of aal1 with rpl1901 mRNA could lead to mRNA decay and global repression of translation. However, this mechanism is not validated experimentally. In addition, a striking conservation of the functional effect of aal1 expression in the fly was demonstrated.

Advance: Strong evidence for a ncRNA acting as a regulator of ageing, which may target a conserved axis of regulation.

Interested audiences: Ageing field / post-transcriptional gene regulation / anyone interested in differences between proliferating and quiescent cells, which has recently drawn increased interest (with much of the published research having been carried out in proliferating cells).

3. How much time do you estimate the authors will need to complete the suggested revisions:

Estimated time to Complete Revisions (Required)

(Decision Recommendation)

Between 1 and 3 months

4. Review Commons values the work of reviewers and encourages them to get credit for their work. Select 'Yes' below to register your reviewing activity at Web of Science Reviewer Recognition Service (formerly Publons); note that the content of your review will not be visible on Web of Science.

No

Review #3

1. Evidence, reproducibility and clarity:

Evidence, reproducibility and clarity (Required)

Large portion of the eukaryotic genome is transcribed producing a number of ncRNAs whose functional significance has remained a matter of intensive research of recent decades. The study by Anver et al., investigate role of ncRNAs in chronological life span in fission yeast.

Anver et al., implicate aal1 ncRNA (SPNCRNA.1530), which is induced in quiescent fission yeast cells in translational control. Authors demonstrate that deletion or overexpression of aal1 can alter polysome profiling where mRNA encoding for ribosomal proteins (in particular rpl1901) are most affected. Overexpression of aal1 ncRNA in yeast or Drosophila extends chronological life span (CLS).

****Comments:****

1. It is not clear whether overexpression of aal1 affects transcription (inferred from RNA seq) or translation of Rs genes in aging cells? This needs to be properly investigated to address the mechanism.
2. Page 6-I am not sure 'dimerization' is a right term for RNA:RNA interaction.
3. Is induction of aal1 ncRNA mediated via transcription or post-transcriptional mechanisms?

2. Significance:

Significance (Required)

Strength - addresses an important question in the field and identifies novel function of ncRNA in CLS using an unbiased approach

3. How much time do you estimate the authors will need to complete the suggested revisions:

Estimated time to Complete Revisions (Required)

(Decision Recommendation)

Between 1 and 3 months

4. Review Commons values the work of reviewers and encourages them to get credit for their work. Select 'Yes' below to register your reviewing activity at Web of Science Reviewer Recognition Service (formerly Publons); note that the content of your review will not be visible on Web of Science.

Yes

Full Revision

Manuscript number: RC--2024-02437

Corresponding author(s): Jürg Bähler

1. General Statements

We thank the three reviewers for their constructive comments. We have carefully addressed the points raised, including additional experiments, analyses, and clarifications in the text, which has improved the manuscript. Below, we provide a point-by-point response to the points raised, with reviewer comments provided in *italics*.

Reviewer #1 (Evidence, reproducibility and clarity (Required)):

This manuscript addresses the function of a lncRNA in yeast that the authors name aal1 (aging-associated lncRNA) that is induced in quiescent cells. They find that deletion of aal1 shortens chronological lifespan, while overexpression prolongs lifespan and these effects are correlated with changes in the levels of ribosomal proteins. Unbiased mapping of epistatic interactions confirms this important functional relationship between aal1 and the ribosome, as does ChIRP-MS approaches that identified RBPs associated with the aal1 RNA. The authors in particular find that rpl1901 mRNA, which encodes a ribosomal protein, is a potential direct binding target of aal1 and the levels of this mRNA are accordingly modulated upon changing the levels of aal1. It is thus proposed that aal1 acts by decreasing Rpl1901 levels, leading to reduced ribosome content that attenuates protein translation and promotes longevity. Amazingly, overexpressing the aal1 lncRNA is sufficient to extend lifespan in flies, despite the fact that aal1 is not evolutionarily conserved in flies. This suggests a conserved underlying mechanism across species that this lncRNA is acting upon. Overall, the authors have done a nice job of using a variety of high-throughput techniques to define a clear molecular mechanism, which they have validated with supporting experiments.

(1) Fig 1C: Please clarify the degree that aal1 is over-expressed in the over-expressed cells.

Reply: As mentioned in the manuscript, *aal1*-pOE cells strongly overexpress *aal1* from a plasmid under the strong *P41nmt1* promoter. We now provide RT-qPCR data in the new Supplemental Fig. 1A, showing an overexpression of 3- to 8-fold (\log_2) relative to empty-vector control cells in stationary phase. We refer to these data on p. 3 when describing the *aal1*-pOE strain used in Fig. 1C.

(2) Fig 2B: To confirm that the overexpressed aal1 transcript accurately recapitulates endogenous localization, the authors should consider performing nuclear cytoplasm biochemical fractionation followed by QPCR to address if endogenous aal1 does shown enrichment in the cytoplasm.

Reply: As pointed out in Reviewer #2's cross-commenting, biochemical nuclear-cytoplasmic fractionation is notoriously tricky in yeast, and this would be a highly challenging experiment. Please note that besides the sm-FISH experiment, our genetic-interaction data, polysome fractionation and polysome profiling experiments, ChIRP-MS and ChIRP-seq results, and *in silico* predictions of RNA interactions all point to a cytoplasmic role of the *aal1* RNA.

(3) Fig 4C: This figure should be zoomed in and more details provided so that the degree of potential base pairing is clear. It is impossible to tell from the data presented and there is minimal description in the text.

Reply: Yes, we have now replaced Fig. 4C with a larger, high-resolution version of the RNA-RNA interaction, which shows the details of the predicted base pairing. On p. 6, we have also added a sentence with further detail on the predicted interaction involving four regions and over 40 base pairs.

*(4) To really prove the base pairing model in Fig 4C, it would be really fascinating to overexpress a mutated form of *aal1* that can not base pair as efficiently to *Rpl1901* mRNA and see if the phenotypes are lost.*

Reply: In principle, this could be an interesting experiment, but it would be very challenging and not necessarily conclusive. The predicted base-pairing between *rpl1901* and *aal1* is extensive and involves several domains (see above). So, extensive mutagenesis would be required to reduce the predicted binding, which would be experimentally tricky to achieve. Moreover, such an extensively mutagenised *aal1* RNA may fail to function for reasons other than reduced binding to *rpl1901*, so this experiment would not really prove the base-pairing model in Fig. 4C.

Minor points:

*(1) Page 3: For completion, it would be useful to mention the known functions, if any, of the *ckb2* and *ctu1* genes that are neighboring *aal1*.*

Reply: Yes, we have now added this information on p. 3. These genes encode a CK2 regulatory subunit (*ckb2*) and a cytosolic thiouridylase subunit (*ctu1*).

Reviewer #1 (Significance (Required)):

This manuscript addresses a novel role for a lncRNA in controlling longevity in yeast and shows that even though the lncRNA itself is not conserved in flies, forced overexpression of it in flies is sufficient to extent lifespan. I think this opens up new avenues for understanding aging related pathways and will be of interest to researchers in both the RNA and aging communities.

Reviewer #2 (Evidence, reproducibility and clarity (Required)):

In their manuscript "Aging-associated long non-coding RNA boosts longevity and reduces the ribosome content of non-dividing fission yeast cells", Anver et al. explore the biological function

of a previously uncharacterised lncRNA, SPNCRNA.1530 (termed aging-associated lncRNA, or *aal1*, based on the findings in this manuscript), which had been identified in a previous study from the same lab as one of four lncRNAs that significantly slowed down growth when overexpressed. *Aal1* levels are restricted by nuclear decay enzymes in proliferating cells, but strongly increase (> 15-fold) in stationary cells. Using a deletion strain and overexpression constructs, the authors demonstrate that *aal1* levels correlate with increased chronological life span and slower growth. This is also true when *aal1* is expressed from an ectopic location, suggesting that the lncRNA functions in trans. *Aal1* localizes to the cytoplasm when overexpressed, and SGA analysis of the deletion strain revealed widespread genetic interaction with genes involved in protein translation. Pulldown of *aal1* with antisense oligos enriched ribosomal proteins and several highly expressed metabolic enzymes, in addition to various ncRNAs and mRNAs. Four of the interacting mRNA encoded for ribosomal proteins, among them *rpl1901* mRNA, for which dimer formation with *aal1* could be predicted. The authors suggest a model where *aal1* modulates levels of ribosomal proteins, potentially through direct interaction with *rpl1901* mRNAs, leading to decreased cellular ribosome content and increased chronological life span. In support of this model, they demonstrate that varying *rpl1901* levels with the help of a tunable promoter also correlated with changes in life span changes. Finally, they also showed that expression of fission yeast *aal1* in the *Drosophila* gut was able to extend life span of the fly.

I raise the following major points:

1) I do not find the data supporting the interaction of *aal1* with ribosomes particularly convincing. Yes, there is sedimentation of the ncRNA on polysome gradients, but the overlap with different ribosomal fractions is not very clear. Also, the pattern of ncRNA sedimentation is very different between wt and OE strain (S4A), whereas the polysome profile shows only little change. Other factors, including condensation of *aal1* in ribonucleoprotein condensates (as the authors suggest) could also influence sedimentation behaviour. Disrupting ribosomes in the gradients and observation of a co-shift of *aal1* could help to support a direct interaction. The ChIRP-MS experiment identified many ribosomal proteins but also many other highly expressed proteins, including *act1*, *adh1* and *pma1*. Setting this in relation to protein expression data could be helpful. However, as I understand it, the model of *rpl1901* repression by direct interaction with *aal1* that the authors suggest does not require an interaction with ribosomal proteins or ribosomes to be functional.

Reply: We agree that the data are not conclusive for a direct interaction of *aal1* with ribosomes. As the reviewer points out, our model does not require such a direct interaction but is rather based on a direct interaction of *aal1* with the *rpl1901* mRNA. However, we provide ample evidence of a functional association between *aal1* and ribosomes based on the genetic interaction assay, polysome profiling data, RNA-seq experiments, and ChIRP-MS assays, and we incorporate these findings into the model presented in the Discussion. Concerning the ChIRP-MS experiments, while several targets are highly expressed proteins, these were not enriched in the ChIRP-MS pull-downs from *aal1*Δ control cells, rendering it very unlikely to reflect technical bias. We did not claim a direct interaction between *aal1* and ribosomes in the

original manuscript and have further tempered the relevant statements in the Abstract and Results.

2) *There must be a better way of quantifying cellular ribosome content than comparing total absorption in polysome profiles. In particular, as the authors state in the methods section "The lysates were quantified in a Nanodrop (OD260). Equal amounts of each lysate were loaded for the polysome fractionation." A260 measurements are dominated by ribosomal RNA. How can this be quantitative? One assay that has been used to determine translation capacity is puromycin labelling (but may not be straightforward in yeast).*

Reply: We agree that the polysome profiling assays are only semi-quantitative, and puromycin labelling is a better method to measure protein translation directly. Therefore, we now performed puromycin incorporation assays to determine the translation capacity of cells that overexpress or lack *aal1* compared to the relevant control cells. These assays showed that translation is reduced when *aal1* is overexpressed, as shown in the new Figure 3I (replacing the old Fig. 3E,F, which were moved to Supplemental Fig. 5B). Cells deleted for *aal1* showed only a subtle increase in protein translation (Figure 3I). This result is similar to the weak effect on cell growth (Figure 1E), consistent with *aal1* being hardly expressed in proliferating cells. These new findings present stronger evidence that *aal1* reduces the capacity for protein translation. We describe the new experiments in the results (p. 5-6) and the Methods (p. 15-16) and mention them in the Abstract and Discussion. In the title, we have replaced 'reduces the ribosome content' with 'reduces the translational capacity'.

Minor points:

1) *Resolution of some of the figures is poor in the submitted pdf. In Fig 4C, this makes it impossible to distinguish both RNAs in the predicted dimer structure. Please check. Also, please use a colour combination different than red and green to make the images accessible to people who are colour-blind.*

Reply: Yes, we have now replaced Fig. 4C with a larger, high-resolution version showing more details of the predicted RNA-RNA interaction and using a colour combination accessible for colour-blind people. We now also provide separate high-resolution files for each figure.

2) *Have the authors run structure predictions with the three other ribosomal protein mRNAs that were detected in ChIRP? Some of these were also very significantly enriched. Could the potential for dimer formation with *aal1* be conserved among these?*

Reply: Yes, we have run these predictions. Among the other three mRNAs encoding ribosomal proteins, only *rpl1802* showed some potential to interact with *aal1*, although less convincingly than for *rpl1901*. We have now added this information on p. 6 and show the predictions' results in the new Supplemental Figure 6B.

3) *At times it is not very clear to me why a particular strain was chosen for a particular experiment. For the SGA analysis, the authors use the deletion strain to test impact on growth under proliferating conditions. Considering that expression of *aal1* is almost fully repressed in*

the wt under these conditions, the impact of the deletion predictably is very low. Using the overexpressing strain (or an assay that tests stationary conditions) might have resulted in stronger or more genetic interactions. Similarly, the differential gene expression experiment is carried out in stationary cells, but here the authors compare the wt (where aal1 is derepressed under these conditions) to the overexpressing strain. Here, comparing to the deletion strain may have rounded off the experiment.

Reply: The power of the genetic-interaction assay is to uncover phenotypes for mutants that do not have a strong (or any) phenotype on their own (such as *aal1Δ*) by pairing them with other mutants. As a complementary assay, we now also used a *aal1* overexpression mutant (*aal1-gOE*), and screened this query mutant for interactions with all non-essential coding-gene deletions in three biological repeats. The 297 genes interacting with the *aal1-gOE* mutant were also enriched in several GO terms related to ribosomes and translation, besides those related to metabolism, mitochondrial function and cell regulation. Thus, genetic interactions for both *aal1* deletion and overexpression mutants point to functions associated with protein translation. We now describe these additional assays on p. 4 and in the Methods, with the data provided in Dataset S1 and the functional enrichments in the new Supplemental Figure 3.

For the gene-expression experiment, we used the empty-vector control as reference cells to compare to the *aal1-pOE* cells, because these strains contain the same genetic background except for the *aal1* overexpression. Note that there is a massive difference in *aal1* expression levels between these two strains, as shown in the new Supplemental Figure 1A.

Referees cross-commenting

Re Reviewer #1's comment (2)

"(2) Fig 2B: To confirm that the overexpressed aal1 transcript accurately recapitulates endogenous localization, the authors should consider performing nuclear cytoplasm biochemical fractionation followed by QPCR to address if endogenous aal1 does shown enrichment in the cytoplasm."

My comment: Biochemical nuclear-cytoplasmic fractionation is notoriously difficult in yeast, so this may be a hard experiment to do.

Re: Reviewer #3's comment (1):

"1. It is not clear whether overexpression of aal1 affects transcription (inferred from RNA seq) or translation of Rs genes in aging cells? "

My comment: In my understanding, the model is that it affects RNA levels, but through a post-transcriptional mechanism.

Reviewer #2 (Significance (Required)):

The link between aal1 expression and chronological life span is well established and strongly supported by the data provided. An involvement in the regulation of protein translation is supported by the genetic interaction data as well as the differential gene expression analysis in the overexpressing strain, and highly plausible in the light of other findings in the ageing field. A possible mechanism is also put forward, suggesting that dimerization of aal1 with rpl1901

mRNA could lead to mRNA decay and global repression of translation. However, this mechanism is not validated experimentally. In addition, a striking conservation of the functional effect of aal1 expression in the fly was demonstrated.

Advance: Strong evidence for a ncRNA acting as a regulator of ageing, which may target a conserved axis of regulation.

Interested audiences: Ageing field / post-transcriptional gene regulation / anyone interested in differences between proliferating and quiescent cells, which has recently drawn increased interest (with much of the published research having been carried out in proliferating cells).

Reviewer #3 (Evidence, reproducibility and clarity (Required)):

Large portion of the eukaryotic genome is transcribed producing a number of ncRNAs whose functional significance has remained a matter of intensive research of recent decades. The study by Anver et al., investigate role of ncRNAs in chronological life span in fission yeast. Anver et al., implicate aal1 ncRNA (SPNCRNA.1530), which is induced in quiescent fission yeast cells in translational control. Authors demonstrate that deletion or overexpression of aal1 can alter polysome profiling where mRNA encoding for ribosomal proteins (in particular rpl1901) are most affected. Overexpression of aal1 ncRNA in yeast or Drosophila extends chronological life span (CLS).

Comments:

1. It is not clear whether overexpression of aal1 affects transcription (inferred from RNA seq) or translation of Rs genes in aging cells? This needs to be properly investigated to address the mechanism.

Reply: As pointed out in Reviewer #2's cross-commenting, our model is that *aal1* affects RNA levels via a post-transcriptional mechanism. This model is supported by the sm-FISH results showing *aal1* in the cytoplasm, by the ChIRP-seq results and *in silico* predictions indicating binding of *aal1* to mRNAs for ribosomal proteins, and by the RNA-seq results showing lowered levels of *rpl1901* when *aal1* is overexpressed. As described in the Discussion, we propose that the interaction of *aal1* with the *rpl1901* mRNA during its translation leads to NMD-mediated degradation of *rpl1901*, a mechanism that has been described for other long non-coding RNAs. This model is supported by the functional associations of *aal1* with ribosomal proteins, based on the results from the genetic interaction, ChIRP-MS, and polysome fractionation assays. We also propose that the *aal1*-mediated repression of the *rpl1901* mRNA leads to the global repression of other ribosomal protein genes (supported by the growth experiments of *rpl1901* mutants), reducing the cellular ribosome content and global protein translation. For the latter, we now provide stronger evidence with the new puromycin incorporation assay in Figure 3I.

2. Page 6-I am not sure 'dimerization' is a right term for RNA:RNA interaction.

Reply: We have now replaced 'dimerization' with 'RNA-RNA interaction' in the main text. Note, however, that the term 'dimerization' is used for this type of interaction, e.g. by the RNAcifold program (<https://www.tbi.univie.ac.at/RNA/tutorial/#sec6>).

3. Is induction of aal1 ncRNA mediated via transcription or post-transcriptional mechanisms?

Reply: As described in the manuscript, we know from our previous studies that *aal1* is targeted by RNA-processing pathways, such as RNAi and nuclear exosome, and becomes upregulated in proliferating cells in RNA-processing mutants (Atkinson et al., RNA 2018, PMID: 29914874). These RNA-processing pathways are downregulated in ageing, quiescent cells (Marguerat et al., Cell 2012, PMID: 23101633), which will facilitate the induction (derepression) of *aal1*. It is well possible, as is often the case with non-coding RNAs, that *aal1* is also regulated at the transcriptional level. However, this manuscript focuses on how *aal1* functions rather than how its expression is controlled.

Reviewer #3 (Significance (Required)):

Strength- addresses an important question in the field and identifies novel function of ncRNA in CLS using an unbiased approach

Dear Prof. Bähler,

Thank you for the transfer of your revised manuscript to EMBO reports. We have now received the enclosed reports from the referees, and I am happy to say that all support its publication now. Referee 3 still has a minor suggestions that I would like you to incorporate before we can proceed with the official acceptance of your manuscript.

A few editorial requests will also need to be addressed:

- Your ms has 5 main figures but separate results and discussion sections. Please either add one more main figure, or combine the results and discussion sections to publish your study as a short report. Short reports can have a maximum of 29,000 characters including spaces but excluding methods and references.
- Please also submit a word file of your ms without figures with your final submission.
- Please add up to 5 keywords to your ms file.
- Please add a "Disclosure and Competing Interest Statement" to the ms file.
- The name Konstantinos Thalassinos is in the ms vs. Kostas Thalassinos in our online submission system, please correct to one name only.
- The reference format needs to be alphabetical (not numerical), et al should be used after 10 author names while DOIs should only be used for preprints and datasets that have not been published yet. The EMBO reports (Harvard) style is also in EndNote.
- Please upload with your final ms a completed author checklist that can be downloaded here: . The completed author checklist will also be part of the transparent peer-review process file (RPF).
- The following FUNDING INFO is missing in our online submission system: Medical Research Council and Bangabandhu Overseas Scholarship, University of Dhaka. Please add this info with the final ms submission.
- There are 4 datasets that need to be renamed to Dataset EV1-EV4; each legend should be provided as a separate tab/sheet in each Excel dataset file (Dataset EV3 has one extra/blank sheet that should be removed).
- The Appendix pdf file needs a title page with title Appendix and a short table of content with page numbers; the correct nomenclature and ms callouts need to be Appendix Figure S1, etc. The separately uplidd Suppl. Table should be part of the Appendix file with the figures, but the correct nomenclature and callouts should be Appendix Table S1. Regarding the supplemental figures, we can accommodate 5-6 EV figures that would then need to be uplidd as individual files per figure and their legends would need to be provided in the ms file, right after the main figure legends (the nomenclature for these figures would be Figure EV1, etc.). You can either chose the most important suppl figures for EV figures, and keep the rest in the Appendix, or leave all figures in the Appendix file. The advantage of the EV figures is that they are imbedded inline in the ms html version and are clickable and expandable. You can also find all information about our file types in our guide to authors online.
- Since July 2024, the Methods section needs to include a Reagents and Tools Table (listing key reagents, experimental models, software and relevant equipment and including their sources and relevant identifiers) followed by a Methods and Protocols section in which we encourage the authors to describe their methods using a step-by-step protocol format with bullet points, to facilitate the adoption of the methodologies across labs. More information on how to adhere to this format as well as downloadable templates (.docx) for the Reagents and Tools Table can be found in our author guidelines: <<https://www.embopress.org/page/journal/14693178/authorguide#manuscriptpreparation>>. If you do not want to re-write the methods, this is OK at this point, but please do send us a Reagents and Tools table.
- Please remove the Supplemental Files section at the end of the ms.
- Please provide the specific URLs for PXD045625 and GSE243036 datasets in the data availability statement.
- Please note that the exact p values are not provided in the legend of figure 4d.
- Please indicate the statistical test used for data analysis in the legend of figure 3a.
- Please note that the box plots need to be defined in terms of minima, maxima, centre, bounds of box and whiskers, and percentile in the legend of figure 3i.

- Please note that information related to n is missing in the legend of figure 4a.
- Please note that the measure of center for the error bars needs to be defined in the legend of figure 1b.

I would like to suggest some minor changes to the abstract that needs to be written in present tense. Do you agree with this:

Genomes produce widespread long non-coding RNAs (lncRNAs) of largely unknown functions. We characterize aal1 (aging-associated lncRNA), which is induced in quiescent fission yeast cells. Deletion of aal1 shortens the chronological lifespan of non-dividing cells, while ectopic overexpression prolongs their lifespan, indicating that aal1 acts in trans. Overexpression of aal1 represses ribosomal protein genes and inhibits cell growth, and aal1 genetically interacts with genes functioning in protein translation. The aal1 lncRNA localizes to the cytoplasm and associates with ribosomes. Aal1 overexpression decreases the cellular ribosome content and inhibits protein translation. The aal1 lncRNA binds to the rpl1901 mRNA, encoding a ribosomal protein. The rpl1901 levels are reduced ~2-fold by aal1, which is critical and sufficient to extend lifespan. Remarkably, expression of the aal1 lncRNA in *Drosophila* boosts fly lifespan. We propose that aal1 reduces the ribosome content by decreasing Rpl1901 levels, thus attenuating the translational capacity and promoting longevity. Although aal1 is not conserved, its effect in flies suggests that animals feature related mechanisms that modulate aging, based on the conserved translational machinery.

EMBO press papers are accompanied online by A) a short (1-2 sentences) summary of the findings and their significance, B) 2-3 bullet points highlighting key results and C) a synopsis image that is exactly 550 pixels wide and 200-600 pixels high (the height is variable). The synopsis image should provide a sketch of the major findings, like a graphical abstract. Please note that text needs to be readable at the final size. Please send us this information along with the final manuscript.

Referee #1:

The authors have addressed all the points I have raised. In particular, inclusion of the puromycin labelling experiment to assess translational capacity further supports their proposed model that aal1 acts through modulating the cell's protein biosynthetic capacity.

Referee #2:

The authors have appropriately addressed my prior concerns.

Referee #3:

While as pointed out by the authors a number of experiments (ChIRP-mass spec, polysome profiling, genetic interaction, smFISH) suggest that aal1 can function in the cytoplasm, the evidence from the experimental data doesn't prove that levels of ribosomal proteins mRNA are controlled at post-transcriptional level, or exclude the possibility of transcriptional regulation. In fact from smFISH presented in Fig 2 it seems that aal1 is also localised to the nucleus. Authors should at least discuss these possibilities

All editorial and formatting issues were resolved by the authors.

Prof. Jürg Bähler
University College London
Genetics, Evolution and Environment
Gower Street
Darwin Building
London, London WC1E 6BT
United Kingdom

Dear Prof. Bähler,

I am very pleased to accept your manuscript for publication in the next available issue of EMBO reports. Thank you for your contribution to our journal.
